# A tridimensional framework for governance in the wildland-urban interface using Pyro-Socio-Ecological Zones

Onofrio Cappelluti[1], Francisco J. Escobedo [2] ✉, Giovanni Sanesi[1] & Mario Elia [1]

Zones where urban land covers meet wildland areas have become a critical point of interaction between human settlements and natural ecosystems at risk of wildfire. These zones span diverse socioeconomic, governance, and ecological dimensions that are often not accounted for with frequently used methodologies. We developed and mapped a tri-dimensional Pyro-Socio-Ecological Zone (PSEZ) framework based on a suite of composite indicators that account for disparate Ecological, Socioeconomic, and Governance contexts in southern Italy (IT) and southern California (CA), USA. In CA, PSEZ with high governance consistently present a lower number of wildfires compared to PSEZ with low governance values. In IT, PSEZ with low governance indicate a lower number of wildfires and more burned area while PSEZ with high governance levels had more wildfires and less burned area. Overall, we found that the distribution of PSEZs and wildfire governance implications differed between CA and IT, underscoring greater spatial fragmentation, disparate contexts, and complex socio-ecological interactions in these peri-urban areas. Findings highlight that care is warranted when applying USA derived methods and concepts to fire-prone, landscapes in different geographies and contexts.

Urban development has led to the expansion of the Wildland-Urban Interface (WUI)[1,2]. This transition zone, where the urban land use covers meet wildland areas, has become a critical point of interaction between human settlements and natural ecosystems[3,4]. Unlike peri-urban areas, which are often a concept that encapsulates the challenges of urbanization, ecological, and environmental impacts, WUI is a fire-centric concept affected by dynamic and complex sets of ecological interactions, land use conflicts, and management practices[5,6]. This zone is also characterized by fragmented ecosystems, increased exposure to invasive species, and heightened wildfire hazard[7,8]. On the contrary, WUI zones also provide carbon sequestration for climate regulation, habitat for biodiversity, recreation and water-related regulating ecosystem services[9,10].

Indeed, identifying, defining, and mapping these areas is often complex due to a variety of issues such as administrative boundaries, land tenure, landscape heterogeneity, and scales[11]. Several studies[12–16]

have analysed the ecological and building morphology characteristics of the WUI. Schug et al. (2023)[15] adopted building density criteria and their location and distance relative to natural vegetation patches for WUI-related mapping, planning, and management purposes (i.e., mapping wildfire risk). For instance, the WUI in the United States of America (USA) is defined according to Glickman and Babbitt (2001)[4] as the area where buildings and wildland vegetation converge, increasing environmental and wildfire risks. Radeloff et al.'s (2005)[14] commonly used definition of WUI refers to areas where structures and wildland vegetation mingle. Furthermore, different European countries' WUI definitions account for differing factors like housing density and tree cover, adjusted to regional characteristics[17].

Governmental organizations are often responsible for resource management, land planning, and emergency response in the WUI, but the socioeconomic, governance, and management dimensions are often not accounted for in the above studies[7,11]. However, governance

[1]Department of Soil, Plant and Food Sciences, University of Bari Aldo Moro, Via Amendola 165/A, Bari, Italy. [2]USDA Forest Service, Pacific Southwest Research Station, Riverside, CA, USA. ✉e-mail: Francisco.Escobedo@usda.gov

plays a critical role since transparent public participation and governmental decision-making processes can effectively improve management outcomes[18,19]. But governance is complex, multi-faceted, often theoretical, and can include various aspects such as transparency, participatory processes, trust, and effective and fair institutions[20,21]. For examples of studies evaluating, measuring, and analyzing for strong or weak wildfire governance structures, please refer to Hamilton et al. (2019)[22], Ager et al. (2017)[23], and Holm and Fischer (2023)[24]. In addition, socioeconomic conditions—reflected by indicators such as the Human Development Index (HDI)—interact with governance to influence both the institutional capacity to manage wildfires and the levels of public trust, civic engagement, and social cohesion needed to address complex socio-environmental challenges in WUI areas[25]. Despite this, governance in peri-urban areas is rarely included in WUI analyses. Including governance systems, in addition to socio-ecological context, in WUI mapping approaches using available and context-relevant data could improve adaptive management in ecologically similar, yet socio-economically different geographies[7,17]. Therefore, there is a need for an integrated approach that considers these multiple dimensions when defining and mapping the WUI in fire-dominated biomes.

Accordingly, this study's aim is to develop a spatially explicit socio-ecological mapping approach and typology to account for governance in fire-prone WUI zones. We do so by integrating ecological, socioeconomic and governance factors in a spatially explicit manner. Specifically, our study has the following objectives:

1. Develop a mapping framework for WUI and peri-urban typologies —or Pyro-Socio-Ecological Zones (PSEZ)—by combining ecological, socioeconomic and governance dimensions.
2. Apply this framework to map PSEZ in different fire-prone Mediterranean WUI ecoregions of southern California (USA) and southern Italy.

Development of such a framework and approach could ultimately help policymakers prioritize conservation efforts and implement more effective fire management and governance strategies to safeguard both ecosystems and human communities within the WUI across disparate regions. Additionally, this will help enhance the resilience and adaptive capacity of these PSEZ against wildfire risk and other socioecological impacts, facilitating more effective and equitable strategies for peri-urban management and conservation across the globe.

## Results and discussion
### PSEZ
Our integrated PSEZ framework can facilitate more informed and targeted land management actions and policies aimed at mitigating wildfire risks and enhancing landscape resilience and ecosystem services[26,27]. The combination of the classified indices into a single map (Fig. 1a, b) highlights the dynamics and distribution among the ecological, social, and governance dimensions, revealing complexity, spatial heterogeneity, and underscoring the need to consider the complex interacting factors related to wildfire management. These results are consistent with previous studies that emphasize the importance of spatial heterogeneity in social-ecological systems[28].

An integrated methodology for mapping the WUI across disparate contexts is crucial to understanding management and resource issues associated with global peri-urban areas. There is an extensive body of literature that has developed WUI typologies and archetypes that account for social vulnerability and risk management related to wildfires[29,30]. For example, Schumann et al. (2024)[31], Rivière et al. (2023)[32], and Paveglio et al. (2015)[29] studied the social vulnerability, participatory mapping criteria, and adaptive capacity related to wildfires. Similarly, Evers et al. (2019)[33] and Wigtil et al. (2016)[30] have developed archetypes of wildfire exposure and identified places in the USA with high wildfire potential and social vulnerability. However,

there are a few typologies that account for governance—*strictu sensu* - in fire-prone WUI zones across disparate contexts.

In terms of accounting for the biophysical dimensions, Radeloff et al.'s 2018[34] pioneering approach of combining building density, fire ember transport distances, and wildland cover to map the WUI has been applied in multiple geographies. A unique aspect of this methodology is the inclusion of a 1 km ( ~ 1.5 mile) buffer distance between individual or clusters of buildings and surrounding wildland vegetation patches, identifying zones potentially at risk from ember-driven fires. However, a critical limitation is that this distance may not be applicable if the vegetation type is not prone to wildfire spread, reducing the accuracy of WUI delineation in such cases. Despite the wide use of this method, it overlooks key socioeconomic and governance dimensions, which are essential for a more comprehensive understanding of the socio-ecological dimensions of the WUI. Modugno et al. (2016)[13] improved WUI analyses by incorporating several European regulatory frameworks and using a broader buffer zone to examine wildfire frequency, offering a more detailed ecological perspective. Nonetheless, this approach neglected key socioeconomic and governance interactions. Schug et al. (2023)[15] further expanded the mapping approach, stratifying global WUI areas based on dominant land covers, including grasslands and other ecosystems, which may or may not be prone to wildfires. They also studied demographic variables and existing biomass; however, they did not include them as variables in the mapping approach, but provide the data as geoanalytics. Bar-Massada et al. (2023)[17] also developed one of the most recent methods for mapping the WUI in Europe, focusing on woody fuels, but without integrating and mapping socioeconomic and governance variables.

Overall, these global-level studies have emphasized vegetation-fuels aspects and building densities while adding socioeconomic and governance variables in separate analyses, thus limiting the ability to comprehensively account for the social-ecological realities of the WUI. These similar methods, first used and applied in temperate wildfire-prone areas of the USA, therefore do not account for different governance, land tenure, and management regimes (e.g., southern Europe, Latin America) nor ecosystems and climates that are not wildfire-prone (e.g., tropical, humid forests)[14,15]. Conversely, our approach integrates these socio-ecological and governance dimensions into a context-relevant mapping framework. The two case studies, applied in different countries and administrative units, but similar ecoregions and climates, suggests a robust and common approach to better identify management and policy needs across varying ecological, socioeconomic, and governance contexts[35–37]. Figure 2 illustrates the variability in both the distribution and diversity in PSEZs between southern California and southern Italy.

Specifically in southern California, the ecoregions are characterized by 24 PSEZs, two of which dominate (i.e., EL-SM-GH and EL-SH-GM), and together cover a substantial portion of the study area (52.76%). In contrast, southern Italy exhibits much more fragmented and diversified ecoregions dominated by 27 PSEZ. The PSEZs with the largest area in southern Italy are characterized by EM-SM-GH (15.26%), EH-SM-GH (14.49%) and EL-SM-GH (9.06%). Thus, the greater array of PSEZ dimensions in Italy reflects a complex landscape characterized by greater diversity in terms of socioeconomic and governance realities (Fig. 2). This variability points to a mosaic of territories and landscapes with diverse human communities and management needs, emphasizing the heterogeneity of the socio-ecological context of southern Italy[38,39].

### Wildfire in the PSEZ
Figure 3a, b display average annual area burned and number of wildfires by PSEZ in both southern Italy and California, respectively. These findings have implications for incorporating socioeconomic and

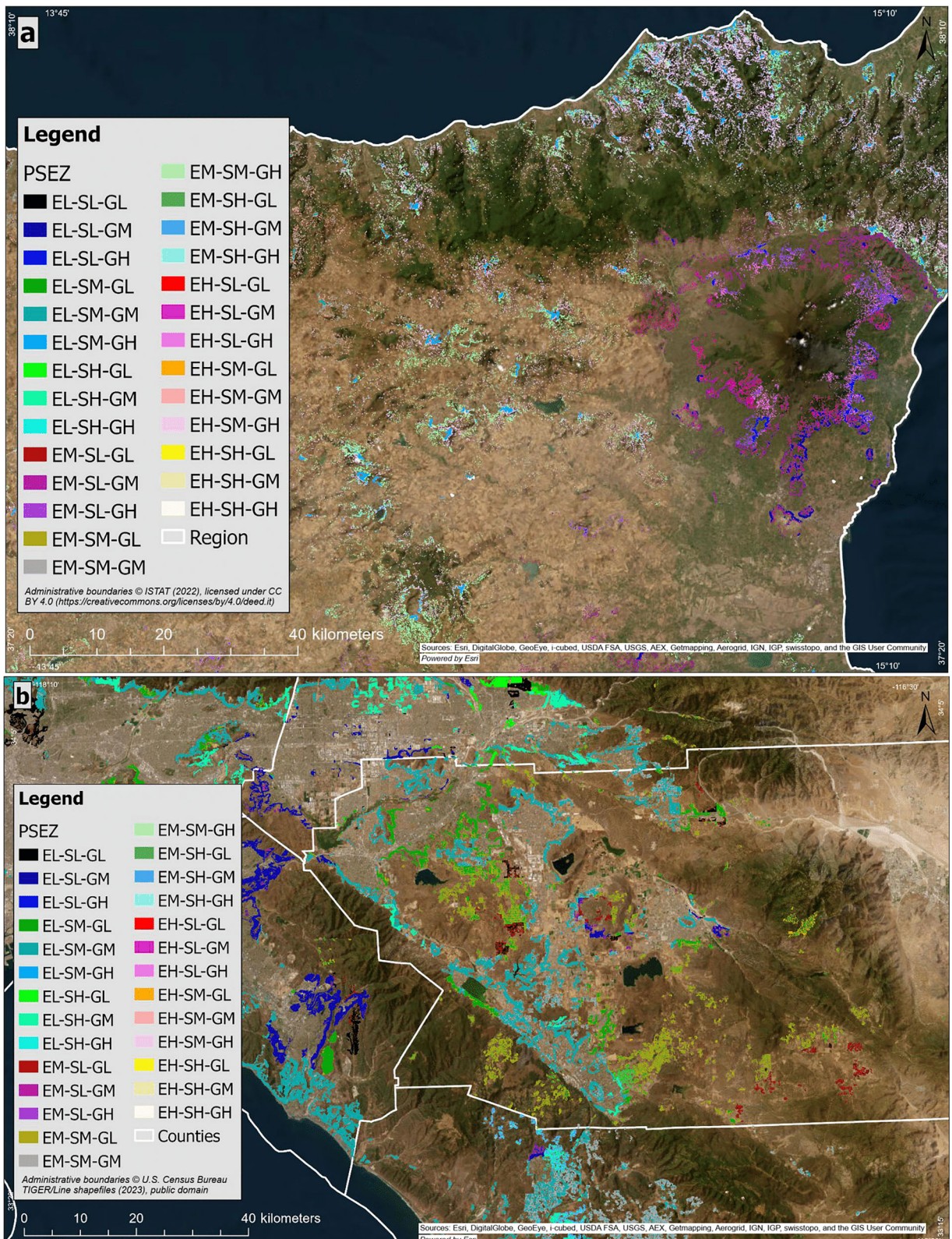

**Fig. 1 | Spatial distribution of Pyro-Socio-Ecological Zones (PSEZ).** PSEZs in southern Italy (**a**) and southern California, USA (**b**). The PSEZ represents the joint interaction of the Socioeconomic Index (S), Governance Index (G), and Ecological Index (E). Each index was classified into three equal ranges, representing low (L) (0–0.33), medium (M) (0.34–0.66) and high (H) (0.67–1.00) values. Administrative boundaries for Italy (Regions) are derived from ISTAT (2022), published under the CC BY 4.0 Attribution 4.0 International License (https://creativecommons.org/licenses/by/4.0/deed.it). Administrative boundaries for the United States (Counties and States) are based on U.S. Census Bureau TIGER/Line shapefiles (2023). Basemap Sources: © Esri, DigitalGlobe, GeoEye, i-cubed, USDA FSA, USGS, AEX, Getmapping, Aerogrid, IGN, IGP, swisstopo, and the GIS User Community; Powered by ESRI.

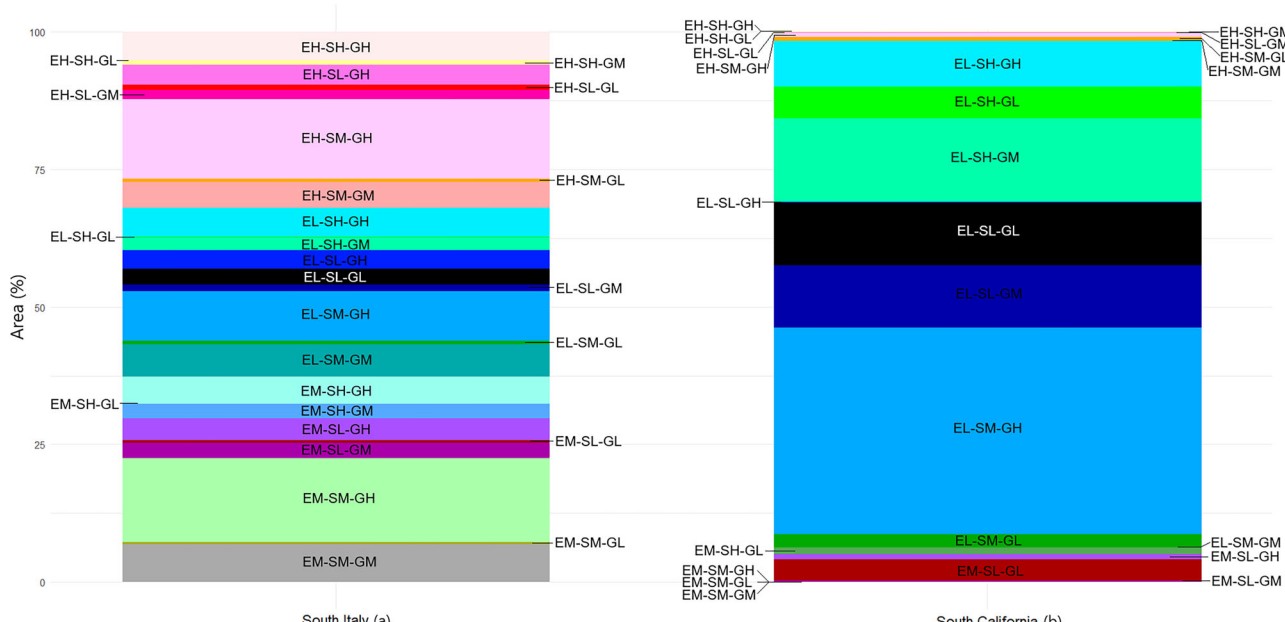

**Fig. 2 | Areal distribution of Pyro-Socio-Ecological Zones (PSEZ).** PSEZ distribution in terms of area (%) in southern Italy (**a**; left panel) and southern California, USA (**b**; right panel). The PSEZ represent the interaction of the Socioeconomic Index (S), Governance Index (G), and Ecological Index (E). Each index was classified into three equal ranges, representing low (L) (0–0.33), medium (M) (0.34–0.66) and high (H) (0.67–1.00) values.

governance dimensions into wildfire management and possibly risk assessments.

We found that in southern Italy (Fig. 3a), PSEZ with high ecological value (EH), present greater burned area (9.76%) and wildfire occurrence (2032.38 events during the analysis period), that are in line with previous studies[40,41]. This suggests that in southern Italy, the distribution of wildfires is less related to socioeconomic or governance factors than in southern California, but likely more related to land cover type and dense vegetation biomass. However, PSEZs showing low governance (GL) indicate a lower number of wildfires over a wider burned area (Fig. 3a) while PSEZs with high governance (GH) have a greater number of wildfires over a lower burned area. Possible reasons behind these contrasting results and high fragmentation of landscape in many PSEZs (Fig. 2), might be the environmental regulatory mechanisms influencing wildfire management and the capacity of local authorities to implement these mechanisms[42]. Italy's decentralized governance structures lead to a range of region-specific guidelines and regulations, differing intra-regional policies, and more localized capacities[43].

Conversely, southern California (Fig. 3b) exhibits marked variability in both the area burned and the number of wildfires across different PSEZs. The greatest area burned occurs in PSEZs with medium ecological values (EM) (4.92%) compared to PSEZs with high ecological values (EH) (3.79%). Similarly, PSEZs with low and medium ecological values (EL–EM) have, on average, a greater number of wildfires than PSEZs (21.20) with high ecological values (EH) (0.95). This result suggests that wildfires are driven by high values of vegetation biomass as well as other numerous variables, as demonstrated by previous studies emphasizing the role of spatial heterogeneity and governance in shaping wildfire dynamics[44,45]. Indeed, fragmented landscapes can contribute to complex wildfire patterns and effects that can be influenced by governance and socioeconomic dynamics[46].

A discernible trend in the number of wildfires shows that PSEZs with high governance (GH) consistently present a lower number of wildfires compared to PSEZs with low governance (GL) values. Also, PSEZs with medium to low governance and socio-economic values show greater wildfire occurrence and burned area, likely due to resource scarcity and a weak governance capacity to manage wildfire in these landscapes. Similarly, cohesive social structures in less

fragmented montane regions often result in more integrated and effective community efforts towards resource and management[18]. Thus, greater governance index values in Italy's South Apennine mixed montane forest ecoregion might stem from more effective policy implementation and enforcement, where less populated, possibly more cohesive populations, have strong local governance structures. Using this same ecoregion, or Southern California Mountains, as an example, the S index values might also imply improved public services and economic stability (Figs. 1, 2).

More specifically, southern Italy's realities differ from those of southern California, where intra-regional and state-level prevention and intervention policies are more uniform due to reliance on federal guidelines and sharing of resources among counties and other administrative units[47]. As such, the PSEZ framework can be useful in capturing the distinct dimensions and spatial mismatches among ecological, socioeconomic, and governance factors that influence wildfire dynamics across disparate contexts. Its application in southern California and southern Italy shows that socio-ecological and governance dynamics in disparate WUIs are indeed shaped by complex ecologies, varying levels of governance capacities, and diverse socioeconomic factors.

We do note that this study has some limitations. First, the availability, quality, and spatial resolution of the datasets will vary considerably across regions, often reflecting differences in methodologies adopted by national and regional governments. In particular, the spatial resolution of socioeconomic and governance data may fail to capture fine-scale heterogeneity within peri-urban or WUI landscapes since the size and configuration of administrative units (e.g., municipalities, census tracts) will differ within and between countries. Second, temporally, PSEZs are dynamic and will change over time in response to ecological changes, socioeconomic developments, and shifts in governance structures; thus, there is a need for long-term monitoring and updates to ensure relevance and accuracy.

That said, our method and approach could be used for future research. First, the data and methods could be used to develop a fourth dimension, like vulnerability or wildfire risk management[12,48]. Although fire risk has been studied and various methods have mapped it[4,9,48,49], governance is different than vulnerability. However, most of

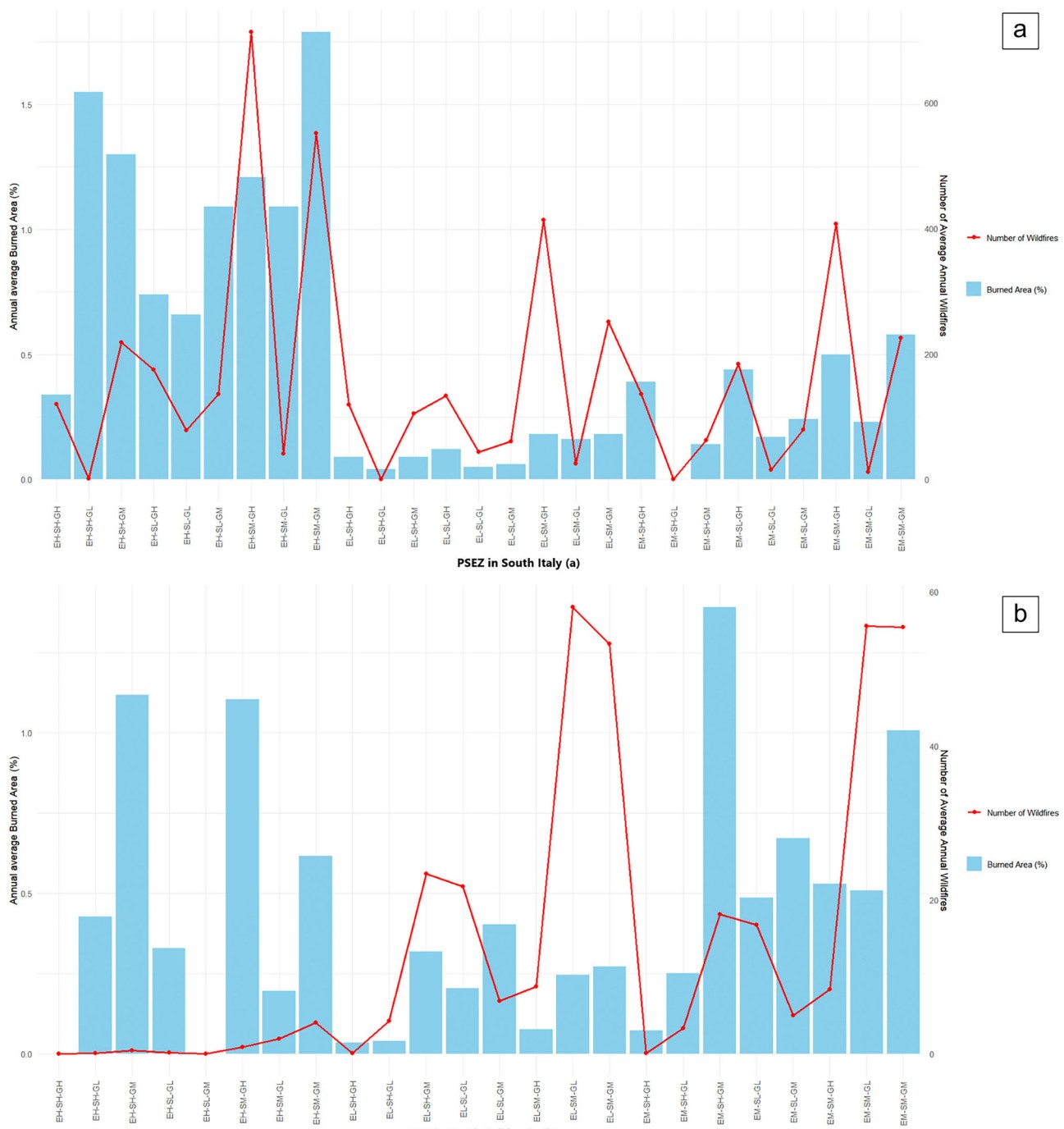

**Fig. 3 | Average annual wildfire occurrence and burned area.** Number of wildfires, and average annual area burned (%), in the same Pyro-Socio-Ecological Zones (PSEZ) in southern Italy (**a**) and of southern California, USA (**b**). The PSEZ represent the dimensional interaction of the Socioeconomic Index (S), Governance Index (G), and Ecological Index (E). Each index was classified into three equal ranges, representing low (L) (0–0.33), medium (M) (0.34–0.66) and high (H) (0.67–1.00) values.

our socioeconomic and governance indicators could easily be incorporated into a framework to account for the exposure, sensitivity and adaptive capacity components of vulnerability[50].

### Implications and implementing the PSEZ framework
The method used in this study builds on well-established approaches in landscape ecology, socioeconomic analyses, and governance literature and provides a globally comparable approach for a comprehensive understanding of fire-prone WUI and peri-urban zones. As such, this study defined and mapped new PSEZ typologies for the WUI by integrating ecological, socioeconomic, and governance

dimensions. These typologies were tested in disparate ecoregions of southern California and southern Italy. We do, however, note that the approach is dependent on the availability of uniform cross-county socioeconomic and geospatial data, which might be temporally mismatched, incomplete, or lacking sufficient spatial resolution.

Despite these limitations, this study's standardized, integrated data sets and our framework represent a policy-relevant approach for better understanding rapidly changing, more frequently fire-affected landscapes across the globe. Our findings highlighted that care is warranted when applying USA-derived methods and concepts to disparate contexts. Similarly, our PSEZ approach could be useful to

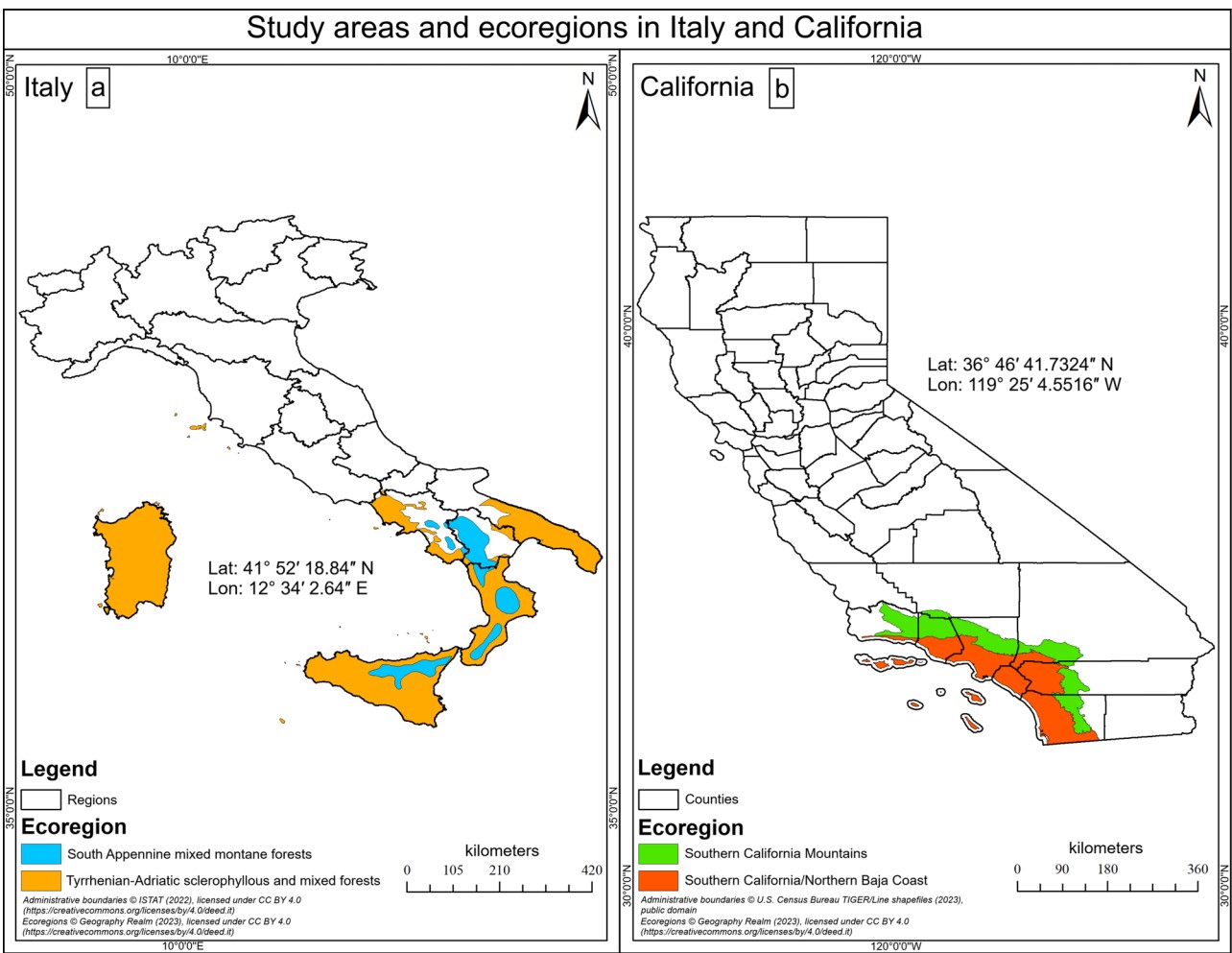

**Fig. 4 | Study areas and ecoregions.** Spatial representation of the study area delineating the two Mediterranean ecoregions from Italy (**a**), and the two ecoregions from California, United States (**b**). Sources for Ecoregions are from the Terrestrial Ecoregions GIS dataset published by Geography Realm (https://www.geographyrealm.com/terrestrial-ecoregions-gis-data/), available under the CC BY 4.0 Attribution 4.0 International License (https://creativecommons.org/licenses/ by/4.0/deed.it). Administrative boundaries (Regions) are from ISTAT (Italy), version 2022, published under the CC BY 4.0 Attribution 4.0 International License (https://creativecommons.org/licenses/by/4.0/deed.it). Administrative boundaries for the United States (Counties and States) are based on U.S. Census Bureau TIGER/ Line shapefiles (2023). The use of all datasets complies with the respective CC BY 4.0 and U.S. Census Bureau public domain attribution guidelines.

identify where fire risk might increase due to greater accumulations of flammable biomass and human-induced ignitions, but can also identify areas where governance systems are weak, communities lack resources, and infrastructure is at risk. This knowledge could be used to target attention on more effective cross-regional public messaging and communications campaigns, fire suppression efforts and fuel management strategies.

Comparing Southern California and Italy revealed spatial differences but, at the same time, similarities in environmental governance and socio-ecological resilience. Local responses to challenges such as land use and climate change pressures are affected by several governance structures and socioeconomic conditions. Accordingly, our PSEZ underscores the need for well-defined management strategies in peri-urban areas and contexts that also align with global sustainability goals. Governance is an increasingly used concept and measure for government and non-governmental organizational transparency, performance and trust. Thus, PSEZ represents a spatially explicit and systematic approach to understanding the interactions occurring in socio-ecological systems and human settlements. As such, they offer an approach for building more resilient, sustainable, and liveable human settlements while reducing conflicts between human activities, resource conflicts, and the effects of global environmental change.

## Methods

**Study areas.** We used publicly available datasets from fire-prone Mediterranean WUI ecoregions in southern Italy and southern California (Supplementary Information Table 1). The ecoregions are highly susceptible to wildfire and are biodiversity hotpots[51]. The ecoregions selected for this study include the South Apennine mixed montane forests (13,094.80 km$^2$ km$^2$) and the Tyrrhenian-Adriatic sclerophyllous and mixed forests (73,406.40 km$^2$) in southern Italy, as well as the Southern California Mountains (15,837.80 km$^2$) and the Southern California/Northern Baja Coast (20,955.50 km$^2$) in southern California (Fig. 4)[52]. The Italian ecoregions encompass the administrative regions of: Calabria, Sardinia, and Sicily, as well as parts of Apulia, Basilicata, Campania, and the islands within the Tuscany region. The Californian ecoregions encompass the Counties of Orange, Ventura, Los Angeles, San Diego, and parts of Santa Barbara, Kern, San Bernardino, Riverside County and the nearby Channel Islands.

### Defining Pyro-Socio-Ecological Zones (PSEZ) framework

We defined and mapped the PSEZ by developing a framework based on a suite of composite indicators that account for an area's Ecological, Socioeconomic, and Governance dimensions (Fig. 5). We then applied this PSEZ framework in two different contexts, but

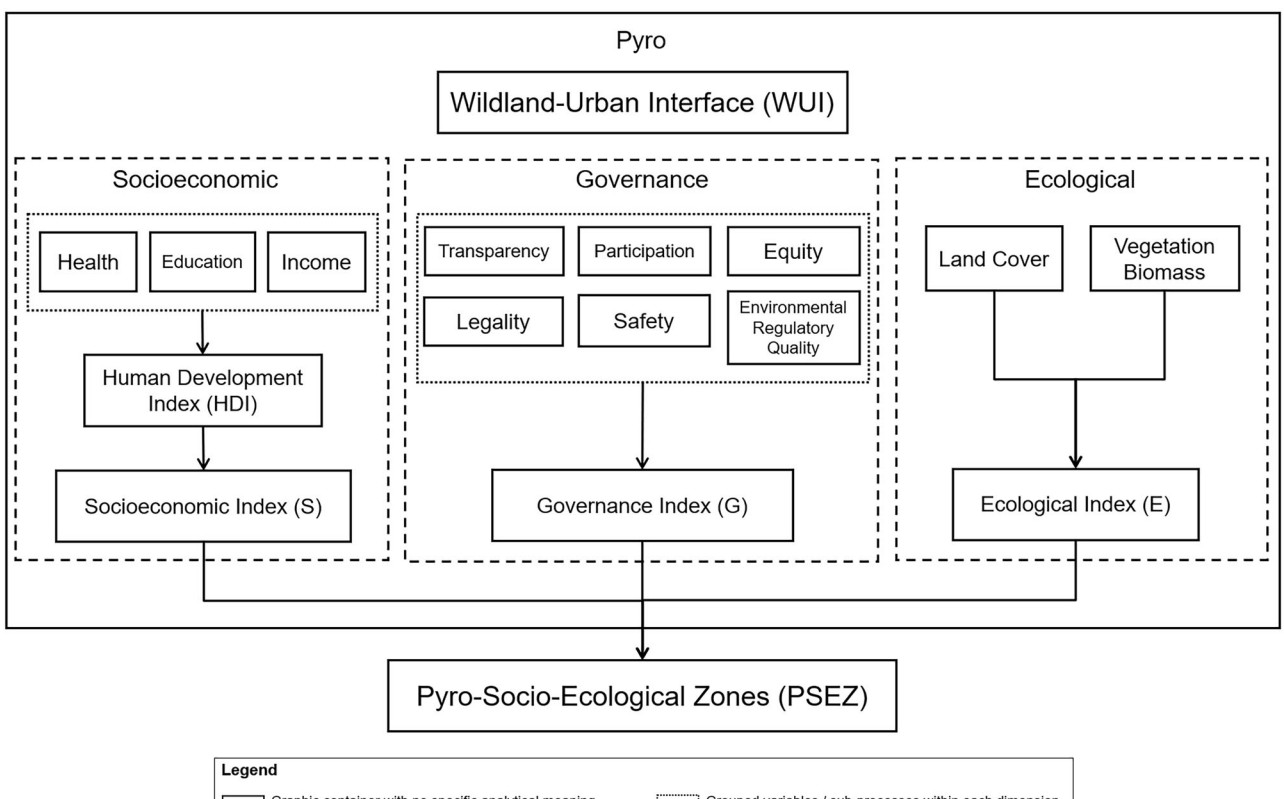

**Fig. 5 | Framework used to develop Pyro-Socio-Ecological Zones (PSEZ) in Wildland-Urban Interface areas and their Socioeconomic (S), Governance (G), and Ecological (E) dimensions.** Dashed boxes delineate the three dimensions and respective indices. Solid boxes indicate sub-processes and indicators. Arrows represent the methodological process for integrating the indices and identifying and delineating the PSEZ.

similar ecoregions[53]: southern Italy and southern California, USA. Developing such a tri-dimensional framework will facilitate the understanding of complex interactions within and among these zones, thereby enabling the identification of spatial patterns, management needs, and policies associated with fire[50]. In the following sections, we outline the development of PSEZs and how we account for the interactions among ecological factors, socioeconomic dynamics, and governance structures that shape these wildfire-prone WUI zones.

### Datasets and definitions

We used the Wildland Urban Interface 2019 dataset from the California Department of Forestry and Fire Protection's Fire and Resource Assessment Program (FRAP)[54] for Southern California that incorporates housing density, fire hazard severity zones, unimproved parcels, and vegetation cover data (Supplementary Information Table 2a, b). For southern Italy, we used the Wildland Urban Interface dataset developed by D'Este et al. (2021)[12], based on Forest Fire Framework Law 2000/353 that provides spatial information for wildfire risk management in Italy (Supplementary Information Table 3).

Socioeconomic dimension (S hereafter) in our mapping framework and typology is based on the Human Development Index (HDI), with values ranging from 0 to 1[55]. The HDI is a globally implemented composite index that measures a country's or region's achievements in three key aspects of human development: health, education, and income[56]. While traditionally used for intra-comparison of countries particularly in the Global South, it has been applied at smaller, more localized scales in places such as Austria, Portugal, Brazil, and Iran[48,57–59]. For southern Italy, we used data from the Italian National Institute of Statistics (ISTAT) that provides comprehensive statistical data on various aspects of Italian society, including demographics, economic, and societal indicators. Specifically, to calculate the Italian HDI, we used the following indicators:

1) Health: measured by life expectancy at birth, indicating the overall health and longevity of a population[60].
2) Education: assessed by mean years of schooling for adults and expected years of schooling for children, reflecting access to and quality of education[61].
3) Income: using Gross National Income (GNI) per capita adjusted for purchasing power parity (PPP) to reflect the standard of living[62].

For Southern California, we used the HDI developed by the Social Science Research Council (SSRC) Measure of America project. The HDI was estimated at the municipal and U.S. Census Bureau Tract scale for Italy and California, respectively (Supplementary Information Table 4).

Governance plays a crucial role in managing complex issues in the WUI[63]. Therefore, governance, our second dimension, was accounted for by using the composite Governance Index (G hereafter), ranging from 0 to 1, and that was based on the 2024 Worldwide Governance Indicators (WGI; Supplementary Information Table 5)[64,65]. The WGI has been used in over 200 countries to account for societal perception of governance quality and is based on six aggregated indicators: Voice and accountability, Political stability and absence of violence/terrorism, Government effectiveness, Environmental regulatory quality, Rule of law, and Corruption control[25,65]. Accordingly, we adapted this framework into 6 indicators based on data availability and their socio-ecological

**Table 1 | The six key indicators used to develop the Governance Index, along with their descriptions and sources**

| Indicator | Description | References |
|---|---|---|
| Transparency | Evaluates corruption levels and ensures accountability within governance systems. | 83–86 |
| Participation | Reflects the extent of participatory engagement and legitimacy of, or trust in, governance processes. | 87–89 |
| Equity | Social spending serves as a key indicator of equity and social justice, influencing societal cohesion. | 90–92 |
| Legality | Identifies the presence of illicit activities that pose a challenge to the rule of law and that erode public trust. | 93–95 |
| Safety | Measures crime rates directly or the level of safety and public order within a society, which are fundamental for maintaining stable governance. | 96–98 |
| Environmental Regulatory Quality | Reflects the implementation of sustainable legal frameworks, ensuring the protection of the environment while maintaining effective regulatory instruments. | 99,100 |

Description of governance indicators were adapted and reorganized from the six Worldwide Governance Indicator (WGI) dimensions[65].

relevance to the two WUI contexts we studied, as shown in Table 1 and Supplementary Information Table 6, and as justified in Supplementary Information Materials & "Methods" Section 1.2.

We defined our third dimension, the Ecological Index (E hereafter), as the combination of land cover categories, their ecological weights, and aboveground biomass density (AGBD) values. The resulting index was normalized to yield a single value ranging from 0 to 1, allowing direct comparison with the other two indices (i.e., the Socioeconomic Index and the Governance Index). This integration of land cover and biomass data assumes that land cover categories differ in the ecosystem services they provide, with these differences captured through their respective ecological weights and complemented by biomass density values. To characterize the ecological dimension in our study areas, we used the National Land Cover Database (NLCD) for southern California and the Copernicus Land Monitoring Service (CLMS) with the Corine Land Cover (CLC) product for southern Italy. The 2021 NLCD dataset offers detailed thematic classifications and metrics at a spatial resolution of 30 meters[66], while the 2018 CLC 2018 dataset, integrated with the National Land Consumption Map 2022 ISPRA, provides high-resolution land use and land cover data at a spatial resolution of 10 meters to align with European Union standards and classifications.

To account for discrepancies between the datasets and enhance their compatibility, we employed a process of aggregation that consolidates the land cover classifications into five comprehensive categories: Forest, Developed, Agriculture Areas, Other Natural Areas, and Other (Supplementary Information Table 7) that are relevant to fire management strategies[67]. Each category encompassed distinct ecological and anthropogenic attributes[68].

An ecological weight was assigned to the different land cover categories (Supplementary Information Table 8) based on studies documenting the ecological impacts of various land cover categories[69–71]. The highest weight was assigned to the forest land cover category, due to the importance for wildland fire management as well as carbon sequestration, biodiversity conservation, and water-related ecosystem services[72,73]. Urban and developed land cover received the lowest weight because of its high imperviousness; however, fires can also occur here, but they are typically wind-driven, building-to-building urban fires rather than wildfires[74,75].

Vegetation biomass is considered a proxy for fuel loadings, ecological productivity and ecosystem service provision (e.g., climate regulation, socio-cultural benefits)[26,40]. We therefore used vegetation biomass type and density data—from remote sensing and forest inventories—to better capture the spatiotemporal dynamics of land-cover change processes and how they interact with socio-ecological outcomes[76]. Specifically, we employed Global Ecosystem Dynamics Investigation (GEDI) data that provides global-level three-dimensional (3D) maps of forest canopies, focusing on canopy height and structural attributes[77,78]. The Level 4B product of GEDI (2021) provides estimates of mean aboveground biomass density (AGBD) at a spatial resolution of 1×1 kilometer[49]. We then normalized the AGBD to yield a single value ranging from 0 to 1.

Since analyses were carried out at the fine scale, we needed to validate the data derived by GEDI. Accordingly, we used the Carbon Monitoring System (CMS) from the Forest Inventory and Analysis (FIA) program[79] and the National Inventory of Forests and Forest Carbon Sinks (INFC[80]), respectively, for southern California and southern Italy. The CMS dataset provided estimates of forest biomass across the conterminous United States based on data collected by FIA from 2009 to 2019. This dataset utilized field survey data from uniformly measured plots distributed throughout the USA, allowing for fine spatial scale biomass estimation. The INFC provided comprehensive data on the quality and quantity of forest resources at both national and regional levels. The INFC, based on data collected from 2005 to 2015, collected several data on forest characteristics and carbon stocks, including quantitative variables measured per hectare and per individual tree or stump[80] (Supplementary Information Table 9).

## PSEZ mapping
The PSEZ were then mapped using the three above-mentioned dimensions and indices (i.e., S, G, and E). To spatially represent PSEZs, we classified the normalized values into three equal intervals representing Low (0–0.33), Medium (0.34–0.66), and High (0.67–1.00) values for each of the three indices. This equal-interval classification method ensures that each class covers an equivalent range, thereby providing a more clear and intuitive understanding of the data distribution[81]. Each index was considered equally important in the analysis and therefore were assigned equal weights in the classification process.

To generate the final PSEZ classification, the spatial layers corresponding to the three indices were intersected in a vector-based environment using GIS tools, allowing the combination of ecological, socioeconomic, and governance information within each spatial unit of analysis. Finally, we used a three-dimensional color matrix to graphically represent the different PSEZ (Fig. 6 and Supplementary Information Table 10). The three axes represent the Ecological (E), Socioeconomic (S), and Governance (G) indices, each categorized into three levels: Low, Medium, and High. This tri-dimensional configuration allows the visualization of all 27 unique combinations of the three indices. Each block within the cube is assigned a unique color, corresponding to a specific PSEZ class, making it possible to intuitively understand how different combinations of ecological, social, and governance conditions are distributed across our study areas. This color-coded matrix was then used to map and compare the PSEZ in both our study areas.

## PSEZ and wildfire occurrence
We applied our matrix (Fig. 6) and analyzed the extent and distribution of PSEZs and calculated the area affected by wildfire in southern California and southern Italy from 2007 to 2022. Specifically, the PSEZ map was intersected with wildfire perimeters to determine average annual burned area and occurrence or the number of wildfires per year. The wildfire extent and occurrence data were from FRAP for southern California and the Command of the Forest, Environmental and Agri-food units (CUFA) for southern Italy.

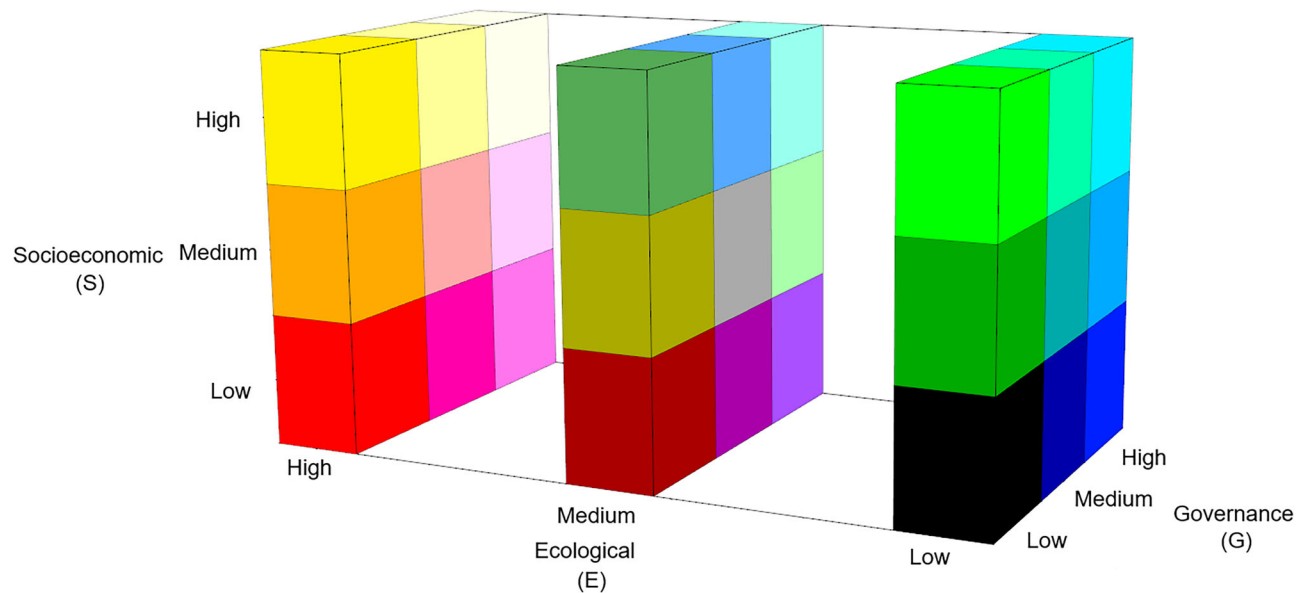

**Fig. 6 | Tridimensional color-coded matrix of Pyro-Socio-Ecological Zones (PSEZs) in southern Italy and California, USA.** Each colored block corresponds to a unique combination of Ecological (E), Socioeconomic (S), and Governance (G) index levels.

## Data availability

Boundary data for ecoregions (Fig. 4) can be downloaded in shapefile format or as a Google Earth Engine dataset from Geography Realm (https://www.geographyrealm.com/terrestrial-ecoregions-gis-data/), based on Dinerstein et al. (2017)[82]. For Fig. 1, Italian Regions are based on administrative boundaries from the Italian National Institute of Statistics (ISTAT; version 2022), available at https://www.istat.it/notizia/confini-delle-unita-amministrative-a-fini-statistici-al-1-gennaio-2018-2/, and United States counties are based on county/state boundaries from the U.S. Census Bureau TIGER/Line shapefiles (2023), available at https://www.census.gov/geographies/mapping-files/2023/geo/tiger-line-file.html. Sources for datasets on the Wildland-Urban Interface (WUI) and Land Cover are listed in Supplementary Information Table 1. Socioeconomic and demographic data, based on the Human Development Index (HDI), are available for California from the Measure of America project (https://www.ssrc.org/programs/measure-of-america/) and for Italy from the ISTAT. Sources for Governance indicators are provided in Supplementary Information Table 6, and sources for vegetation biomass data are detailed in Supplementary Information Table 8. Wildfire extent and occurrence data were obtained from CAL FIRE's Fire and Resource Assessment Program (FRAP) for southern California, available at https://catalog.data.gov/dataset/california-fire-perimeters-all-b3436/resource/6955eaf7-6452-4922-bc7d-bdac9091c538?inner_span=True, and from the Command of the Forest, Environmental, and Agri-food units (CUFA) for southern Italy, available at https://www.dati.gov.it/view-dataset?tags=aree-percorse-dal-fuoco&page=1. Basemap sources for Fig. 1a, b are from ESRI, Maxar, Earthstar Geographics, and the GIS User Community. Shapefile data for the Pyro-Socio-Ecological Zones (PSEZ) (Fig. 1a, b) are available upon request (contact: onofrio.cappelluti@uniba.it).

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

## Acknowledgments

This project was partially funded by the Wildland-Urban Interface- Fire Effects on Ecosystem Services Focused Initiatives project and annual research appropriations from the USDA Forest Service- Pacific Southwest Research Station (F.J.E.). The study received support and funding from the: 1) Agritech National Research Center and the European Union NextGenerationEU (PIANO NAZIONALE DI RIPRESA E RESILIENZA (PNRR) – MISSIONE 4 COMPONENTE 2, INVESTIMENTO 1.4–D.D. 1032 17/06/2022, CN00000022) and 2) National Recovery and Resilience Plan, Mission 4 Component 2 Investment 1.4 - Call for tender No. 3138 of 16 December 2021 of Italian Ministry of University and Research, funded by the European Union - NextGenerationEU; Project code: CN00000013, Concession Decree No. 1031 of 17 February 2022 adopted by the Italian Ministry of University and Research, CUP: H93C22000450007, Project title: National Center for HPC, Big Data and Quantum Computing (O.C., M.E. and G.S.). This manuscript reflects only the authors' views and opinions; neither the European Union nor the European Commission can be considered responsible for them.

## Author contributions

O.C. and F.J.E. designed research; O.C. and F.J.E. performed research; O.C. and F.J.E. developed the framework; O.C. analyzed data; M.E. and G.S. edited & reviewed the paper; G.S. supervised the research; O.C., F.J.E., and M.E. wrote the paper.

## Competing interests

The Authors declare no competing interests.
