## [Transparent Peer Review File · Nature Communications]

A Tridimensional Framework for Governance in the Wildland-Urban Interface Using Pyro-Socio-Ecological Zones

Corresponding Author: Dr Francisco Escobedo

Version 0:

Reviewer comments:

Reviewer #1

(Remarks to the Author)

This well-written paper presents an approach to understanding specific conditions of WUI areas. In combining indicators of socio-economic, governance, and ecological factors, the pyro-social-ecological zones presented here provide WUI archetypes that have practical application to tailoring risk mitigation in these areas. I can see the clear value for this kind of work on a practical level, and the push in the literature towards this kind of work. The specific approach used in the paper is sound, the data used is appropriate, and it is generally very well presented. I've included some comments below. Very few major comments, most are minor line number comments. Overall I really like the idea presented here, and it's a well-done paper.

1) A more major comment is that the paper neglects to mention the wide array of literature that covers social vulnerability in the WUI, and other WUI typologies/archetypes. There is quite a bit of work out there that is highly relevant to include in this manuscript (and in fact, if omitted could mislead the reader that this is the seminal work on the topic). I would suggest that the authors include this on p. 3 where WUI mapping is discussed, and on p. 13 (probably a whole paragraph needed there). Some papers/authors to consider including: Wigtil et al. 2016, Schumann et al. 2024, lots from Travis Paveglio, Sandra Oliveira, Riviere et al. 2023, Alan Murray, Evers et al. 2019 etc.

2) Discussion of the limitations of your data, and implications they have on your results needs to be expanded.

3) Throughout I've noted some issues in the line numbered comments about difficulty interpreting the many colors for all the crossed categories for the three variables. I've included some specific suggestions that might help with some of the issues, but the reader will still be challenged to interpret the details with so many colors (even those with no accessibility limitations such as color blindness). Wondering if the authors considered any other presentation approaches, or was this the most successful? Its quite a difficult data visualization challenge to be honest.

4) A mention of future work in your conclusion/end of discussion would be a welcome addition. Are there plans to extend the work to more broad regions? What are the challenges there? Any plans for practical application and extension to matching risk mitigation actions?

Line comments:

Abstract: I would suggest some mention of the primary results of your two study areas PSEZ distributions and how they contrast.

Line 22: for your keywords, consider adding "wildland fire" or "wildfire"

Line 38-42: finding this awkwardly presented. Maybe focus on one reference at a time, or make it a general statement with the numbered citations included.

Line 69: I believe Nature Communications puts the Methods section at the end? If this requires re-organization, some re-

wording will be required.

Line 88: reference needed for point made re “similar ecoregions”

Figure 2: really like the overview here, very clear. I'd suggest for all acronyms in the image of the figure, to include both the full name and acronym.

Line 107: no hyphen needed between “demographic-characteristics”

Line 128-129: this sentence probably applies to everywhere with a WUI problem. Suggest rewording, or be more specific and cite something.

Line 130: reference 33 and 34 seem like odd choices at a glance, consider changing these to something more relevant?

Line 134-136: How up to date is this data? Also, this kind of limitation should be mentioned in discussion, and discuss how it can impact your results. Especially relevant in areas with frequent fire regimes resulting in changed fuels, or areas with fast paced land use changes. Also, particularly important given the dynamic political situation in the US...

Line 159-160: Think about rewording this sentence, as it isn't impervious to fire, it just becomes an urban conflagration not a “wildfire”, plus not necessarily low fire occurrence either.

Line 165-167: is this a quote directly from reference 66? I'd suggest just rephrasing and citing.

Line 167: acquisition date for GEDI data?

Line 178: acquisition date for INFC data?

Figure 3: I don't find this figure to be very well formulated. It's maybe just a bit clumsy the way two variables are combined first, and then the third is added. Presenting multivariate data in color has a lot of challenges, but spending some time on getting something presented here that is more intuitive for the reader is strongly recommended. I thought a bit about something to suggest – and I believe a RGB color space cube approach might work for you. You'd have G on the x axis, E on the y axis, and S on the z axis. I've attached a sketch. You could probably keep the existing color scheme (and not have to redo all your maps etc!). Displaying this legend as a 3D shape would be more intuitive than the existing presentation.

Line 215: I think Nature Communications keeps results and discussion as two separate sections? Re organization required if so.

Line 217-219: I find this an out of place sentence for the start of the results

Figure 4: I find the maps quite difficult to decipher with so many colors. I don't have any great ideas as an alternative though. Also, if you do re-work Figure 3 as I've suggested above, you could use the cube in place of the legend?

Page 13: this section where you're discussing WUI delineation and the limitation that they don't include socioecon/governance factors – as mentioned in my main comments, I'd add a paragraph here that focuses on studies that look at WUI typologies/archetypes or social vulnerability in the WUI.

Figure 5: you could consider removing the legend, and just label the zone names directly on the bars in the figure. Would mean the reader wouldn't have to attempt to match colors. I'd probably put the labels between the two bars and just use arrows to point to them.

Line 271-272: remove the sentence that starts with “These classes reflect...”, not necessary here.

Line 276-277: not sure “interact more intricately” is the right wording. Its not that they have a more intricate interaction, just more diversity in the categories..

Line 281-282: remove first line, not necessary

Line 293: Figure 6a is Italy though? Also, throughout the manuscript, keep the order of presentation consistent (in text, and in figures) i.e. always do Italy first, or always so Cali first.

Line 324-327: this sentence is circular, doesn't make sense

Line 334-336: limitations should be included in discussion somewhere, apologizes if I've missed it there.

Line 354: change to “...activities and the effects of global environmental change.”?

Supplementary Material

Supplement table 5: do the authors have any concerns about some of the news sources used for corruption cases?

Reviewer #2

(Remarks to the Author)

The premise of this paper is very interesting and compelling. Information about governance would greatly advance spatial data mapping tools indicating the need, potential, and opportunity for wildfire risk reduction in populated fire-prone areas.

However, this paper does not provide a comprehensive review of the relationship between governance and its various dimensions and wildfire risk and wildfire risk management. The article also does not describe the data used to create the six indicators, how they were constructed, and precisely what they mean.

The article uses a Human Development Index to capture the "Socioeconomic and demographic dimension," but it is explicitly stated for what purpose, evidently to indicate social vulnerability and the need to reduce exposure?

The authors provide more detail about the construction of the pyro and ecological zones. However, how new constructions of these spatial data layers improve upon previous efforts is unclear. The pyro zone seems to be a new name for existing WUI data layers.

Organizationally, the article would have benefited from separating the Results from the Discussion and providing a clear report of results for the overall PSEZs, each zone, and the various indexes. Then, the discussion should focus on the improvements and limitations of these new spatial tools in relation to existing tools and conceptualizations of fire-prone geographies.

Reviewer #3

(Remarks to the Author)

General comments

The topic is interesting.

The text is well written with some typos (e.g., than instead of then in line 311; mention to use the US and after use USA)

The relevance of the methodology and the effective impacts of its practical utilization must be emphasized.

Detailed comments

Title

The focus is not only on governance but on wildfire management. Should the title shift to Pyro-1 Socio-Ecological Zones: A Tri-Dimensional Framework for wildfire management in the Wildland-Urban Interface?

Abstract

Line 12, you did not consider the ecological dimension and inserted the management that is not considered in the framework.

In the keywords, you consider Land use change and Land cover change, although you just considered land cover in the framework. In addition, there is no mention of the temporal shift in land cover. What is the relevance of considering peri-urban areas?

Introduction

How do you define WUI and peri-urban areas? How did you spatially define the WUI area? And the peri-urban areas? How did you delimit the WUI?

Line 62-67 – This statement is interesting. You must demonstrate how your framework can be used to enhance wildfire management effectiveness, improve governance, build resilience, and adaptive capacity.

Methodology

What is the dimension of the pixel used in the maps? For biomass, the spatial resolution of 1 x 1 kilometre

What is the relevance of each of the variables selected?

How can the framework help to identify management needs and areas of vulnerability?

How can you obtain Gross National Income (GNI) per capita values adjusted for purchasing power parity at the pixel level?

In the pyro component of the model, you use WUI. WUI is the area where you are going to apply the model. Thus, it should be out of the model. In the pyro component, wildfire characteristics could be another option, or remove pyro from the model as a component

How can you get the data of the governance indicators at the pixel level?

The weight attributed to the indicator should be considered in a table in the text

In Figure 2, the three dimensions are pyro, socio, and ecological; however, in the text, you mention S, G, and E. What about removing pyro and separating socio into socio and governance?

Results and discussion

You state "Our integrated PSEZ framework facilitates more informed and targeted land management actions and policies aimed at mitigating wildfire risks and enhancing landscape resilience and ecosystem services". Please justify how this can be obtained; otherwise, it is a general statement.

Line 308 – What is the unit of the value (2032.38)?

You state that “the PSEZ framework can be useful in capturing the interactions among ecological, socioeconomic, and governance factors that influence wildfire dynamics across disparate contexts.” In my point of view, you do not analyse the interactions between ecological, socioeconomic, and governance factors.

You state “ Its application in southern California and southern Italy shows that socio-ecological and governance dynamics in disparate WUIs are shaped by both ecological dimension, local governance capacities, and socioeconomic factors.” In my point of view, this is a round explanation. It is the first time you mention local governance capacities. What is the meaning?

Conclusion

Emphasize clearly the strengths and limitations of the work.

Version 1:

Reviewer comments:

Reviewer #1

(Remarks to the Author)

The authors were very detailed and mindful in their revision and responses to the review comments. Excellent job, it's a really nice paper. I think this manuscript is ready to go, short of a few very minor edits. I've got some noted below, by line number:

Line 30: WUI, not WUIs

Line 32: throughout paper, remove the extra spaces before the superscript references and make sure format matches that of the journal requirements

Line 50: WUI, not WUIs

Line 72: Maybe change to “Development of such a framework and approach could ultimately help...”

Line 83-87: flip order so it is always ITA then Cali

Figure 2: use full name (“Socioeconomic”) on the box currently labeled just “Socio” to match the other 2 dimensions. Figure is much improved from last version

Line 118: the subsection heading 2.3.2.1 is probably not necessary?

Line 119: “and demographic” included in S? If it is socioecon and demographic, then that needs to change in Fig 2 and in text above where S is referred to as socioecon

Line 123: “cross-country” is a bit ambiguous, maybe use “intra-comparison of countries”?

Line 136: remove link here, just cite in references

Line 139: this section should be 2.3.3 I think? Then adjust following sections accordingly

Line 140: “Governance plays a crucial role in managing complex issues in the WUI40. Therefore” is not necessary here. Start sentence with “Governance, our second...”. If you'd like to include this point, maybe move to intro where governance is discussed?

Line 143: Why is ref #25 not a reference to the source data for the WGI? Also look at formatting for the ref here, maybe try “(WGI; Supplement Table 5)25.”

Line 143-145: you probably do not need to include the original list of indicators from WGI, and could get away with just saying that your indicators are based on theirs, but modified based on what data you had access to that could represent those indicators? Or any other justification on their selection. I was confused looking at the list here vs the table that they didn't match and had to re-read for clarity. And I'm not sure why ref #41 is here, it doesn't mention any of these indicators.

Line 150: change numbering of section to 2.3.4, and I'd suggest re-naming the section to “Ecological Index” and then move lines 192-199 here (with some slight re-wording for flow). Then use the subheadings for “Land Cover” and “Vegetation biomass” to follow the paragraph on the ecological index. I think that would flow better, and also match the structure of the sections for the other indices.

Figure 3: yes glad this presentation worked out, looks good!

Line 235: change "highlights" to "highlight"

Line 284: remove "The"

Fig 5: heading at the top of the figure image is not necessary, remove and rely on the figure caption below.

Line 293-300: I just realized it would be nice to add in some descriptive statistics on the diversity to go along with your description here. Maybe a pure count of how many zones each region has (i.e. "richness"), and then maybe something like the Simpson's index, or relative diversity/evenness?

Line 302: change "for the same" to just "by"

Line 325: change to "Conversely, southern California..."

Line 330: change to "This result suggests that..."

Line 339: the start of this paragraph doesn't flow from the previous

Line 358: reference format needs adjusting

Line 377: change "in" to "identify"

Line 378-379: maybe change to "...are at risk. This knowledge could be used to target attention on more effective..."

Line 382: change "like" to "such as"

Line 383-384: not sure the wording here is right. Maybe merge the point with your point at the end of the paragraph?

Supplementary materials: I'd suggest that anywhere possible, try to condense text in the tables, use point form (e.g. the descriptions in Table 1)

Suppl Line 2: capitalize "Socio"

Reviewer #2

(Remarks to the Author)

The authors made some effort to revise the paper to address the reviewers' comments. However, the changes were limited. The paper still does not provide a coherent conceptual model of governance and how it relates to wildfire risk or how it could be managed more effectively. While the indicators intuitively seem important, the authors do not explain their relevance. The authors did little to address my previous comments about this shortcoming, and the few sentences they added simply state that the constructs are important, not why. Additionally, the authors statement that the study did not aim to address fire risk and its management was perplexing. At the end of the introduction, the paper explicitly states that the framework will help policymakers implement more effective fire management and governance strategies. Incidentally, the paper never explains how the framework will do this. Revisiting the claim about this implication would have been helpful in the discussion.

Again, the article uses a Human Development Index to capture the "Socioeconomic and demographic dimension," but the authors do not explain the relevance of the indicators to the prioritization of conservation efforts or the implementation of more effective fire management and governance strategies to safeguard both ecosystems and human communities. The added sentence about socioeconomic conditions interacting with governance quality was general and somewhat vague. I don't see any edits to the section on socio-eco-demographic characteristics, so it appears that the authors were not able to provide a clear explanation of the relationship between the HDI indicators and governance and wildfires.

The authors still do not explain how the ecological index improves upon existing spatial wildfire risk data products. While the authors provided an interesting discussion of the novelty of their framework in their rebuttal document, I see very few edits to that section of the paper. Incorporating some of the rebuttal comments into the revised manuscript would have helped.

While the authors tried to address the limitations of their study, they did not expand the discussion of the findings or how their framework could improve upon existing tools and conceptualizations of fire-prone geographies. They could have discussed questions such as: What are the implications of the various combinations of governance, HDI, and ecological index values for wildfire risk, and fire management and governance strategies? Why would various levels of the governance indicators be associated with the number of wildfires and burned area? In fact, some of the additions to the Discussion introduced more questions than answers. How could the framework be used to "develop a fourth dimension like vulnerability or wildfire risk management"? How could the socioeconomic and governance indicators account for wildfire exposure?

Reviewer #3

(Remarks to the Author)

Dear authors,

Thank you for your replies.

Point-by-point response to the reviewers' comments:

Reviewer #1 (Remarks to the Author):

This well-written paper presents an approach to understanding specific conditions of WUI areas. In combining indicators of socio-economic, governance, and ecological factors, the pyro-social-ecological zones presented here provide WUI archetypes that have practical application to tailoring risk mitigation in these areas. I can see the clear value for this kind of work on a practical level, and the push in the literature towards this kind of work. The specific approach used in the paper is sound, the data used is appropriate, and it is generally very well presented. I've included some comments below. Very few major comments, most are minor line number comments. Overall I really like the idea presented here, and it's a well-done paper.

1) A more major comment is that the paper neglects to mention the wide array of literature that covers social vulnerability in the WUI, and other WUI typologies/archetypes. There is quite a bit of work out there that is highly relevant to include in this manuscript (and in fact, if omitted could mislead the reader that this is the seminal work on the topic). I would suggest that the authors include this on p. 3 where WUI mapping is discussed, and on p. 13 (probably a whole paragraph needed there). Some papers/authors to consider including: Wigtil et al. 2016, Schumann et al. 2024, lots from Travis Paveglio, Sandra Oliveira, Riviere et al. 2023, Alan Murray, Evers et al. 2019 etc.

Thank you for this comment and you are correct. Ours is not a seminal paper on the above topics because there are several other WUI risk management and vulnerability typologies/archetypes articles out there. Further, we used the terms “vulnerable/vulnerability” unnecessarily and in several instances throughout our manuscript. However, “vulnerability” is a very different concept than “governance”, so our aim and approach is very different than the references you mention^{1,2}. So, we now see that this led to confusion and in fact we should have avoided excess use of the terms “vulnerability” and “vulnerable” since we were not studying this concept. Also, we should not have made any direct inferences that our approach could be used to directly “map fire risk”. So, we have addressed these shortcomings in two ways.

First, we have replaced vulnerable/vulnerability with the terms: “policies, management, resource needs, and human communities” where appropriate throughout the manuscript to better align with our study’s aim. In fact we have slightly modified our study aim to now read, “.... this study’s aim is to develop a spatially explicit socio-ecological mapping approach and typology to account for governance in fire-prone WUI zones.” Also, since ours was not a mapping typology for exposure to, or management of, risk like those of Evers et al., 2019³, Rivière et al., 2023⁴ or Wigtil et al., 2016⁵; we made sure that “risk management” implications were only mentioned twice. Once when referring to a published fire risk management article⁶ and another use in our future research section.

Second, we address your comment in our response to page 13 by adding a new paragraph where we acknowledge the above and succinctly summarize many of the studies you pointed out (we do note that the journal has a reference limit). That said, please refer to Question #1 and this new paragraph to see the specific response to your concern and how it was addressed in our revisions. But basically, we acknowledge the specific aim and justification for our manuscript and then we state how this approach builds off of these other articles you indicate, specifying that these are

about vulnerability to, and risk management of, wildfires. Again, please refer below to our responses to your concerns below in Question 13.

2) Discussion of the limitations of your data, and implications they have on your results needs to be expanded.

Thank you for the suggestion. We have now, in a new paragraph expanded the final part of the Discussion section to clearly address the main limitations of our study, including differences in data availability, resolution, and methodology across regions, and the implications these may have on the interpretation of our results. Please refer to the last paragraph of the revised Results-Discussion for this new text.

3) Throughout I've noted some issues in the line numbered comments about difficulty interpreting the many colors for all the crossed categories for the three variables. I've included some specific suggestions that might help with some of the issues, but the reader will still be challenged to interpret the details with so many colors (even those with no accessibility limitations such as color blindness). Wondering if the authors considered any other presentation approaches, or was this the most successful? Its quite a difficult data visualization challenge to be honest.

We sincerely thank the reviewer for the thoughtful comments and suggestions regarding the data visualization. We agree that representing multiple cross-categorical variables in a single figure poses a considerable challenge.

So, to address the reviewer's recommendations, we have carefully revised the figure to improve clarity and interpretability. The revised version now directly incorporates the Reviewer's specific suggestions (thank you!), and we believe it now offers a more user-friendly and effective visual graphic (Figure 3). Moreover, to further support readers who may wish to explore the data in greater detail, we are happy to make the PSEZ shapefiles available upon request. This solution should allow any user to access and interact with the spatial data directly, enabling a more personalized and comprehensive interpretation of the results.

4) A mention of future work in your conclusion/end of discussion would be a welcome addition. Are there plans to extend the work to more broad regions? What are the challenges there? Any plans for practical application and extension to matching risk mitigation actions?

Thank you for this suggestion. At this point there are no plans – or funding - to extend the work to other dimensions or regions. However, we now include new text in the form of a future research paragraph that states, “That said, our method and approach could be used for future research. First, the data and methods could be used to develop a fourth dimension like vulnerability or risk to wildfire (12, 34). Although fire risk has been studied and methods have mapped it (4, 9, 34,75), governance is different than vulnerability, however most of our socioeconomic and governance indicators could easily be incorporated into a framework to account for the exposure, sensitivity and adaptive capacity components of vulnerability⁷. Second, our PSEZ approach was used for the WUI and fire, but it could also be adapted and applied to peri-urban areas where issues such as ecosystem service provision, conserving habitat, and formulating land use policies near human settlements are more relevant than fire (76).” We hope this addresses your comments.

Line comments:

Abstract: I would suggest some mention of the primary results of your two study areas PSEZ distributions and how they contrast.

Thanks for the comment. We slightly revised the last 2 sentences of our abstract added new text in this revised manuscript's abstract according to results across the two study areas and their PSEZ distributions and how they contrast. The revised 2 sentences read, "Overall, we found that the distribution of PSEZs and wildfire governance implications differed between CA and IT, underscoring greater spatial fragmentation, disparate contexts, and complex socio-ecological interactions in these peri-urban areas. Findings highlight that care is warranted when applying USA derived methods and concepts to fire-prone, landscapes in different geographies and contexts."

Line 22: for your keywords, consider adding "wildland fire" or "wildfire"

Done. We have added "wildfire management" to the list of keywords. The 'management' term was added so as to address a comment by another reviewer as you will see later.

Line 38-42: finding this awkwardly presented. Maybe focus on one reference at a time, or make it a general statement with the numbered citations included.

Thank you for the suggestion. We revised the sentence to improve clarity and followed your recommendation. The updated version now reads more clearly.

Line 69: I believe Nature Communications puts the Methods section at the end? If this requires re-organization, some re-wording will be required.

Thank you for your comment. While Nature Communications typically places the Methods section at the end of the manuscript, this is not mandatory. As stated in the author guidelines: "Methods are typically placed at the end of the article before the References." Given that our study is methodological in nature, we intentionally placed the Methods section earlier to provide the necessary background to the methodology and improve clarity for the reader. That said, if requested by the editors, we are happy to move this section to the end of the manuscript.

Line 88: reference needed for point made re "similar ecoregions"

Thank you for your observation. A reference has now been added (i.e., Olson & Dinerstein, 2002).

Figure 2: really like the overview here, very clear. I'd suggest for all acronyms in the image of the figure, to include both the full name and acronym.

Done. We followed your suggestion and added both the full names and acronyms in the figure.

Line 107: no hyphen needed between "demographic-characteristics"

Thanks for catching the error. The hyphen has been removed as suggested.

Line 128-129: this sentence probably applies to everywhere with a WUI problem. Suggest rewording, or be more specific and cite something.

Thank you for the suggestion. The sentence has been rephrased to be clearer. It now reads, "Governance plays a crucial role in managing complex issues in the WUI"⁴⁰

Line 130: reference 33 and 34 seem like odd choices at a glance, consider changing these to something more relevant?

We agree that references 33 and 34 might not be the most appropriate sources to support our point regarding the importance of context-specific governance in fire-prone regions like Southern Italy and Southern California. In response, we have replaced these with a more relevant reference:

Fischer et al. (2016)⁸, Wildfire risk as a socioecological pathology (Frontiers in Ecology and the Environment).

Line 134-136: How up to date is this data? Also, this kind of limitation should be mentioned in discussion, and discuss how it can impact your results. Especially relevant in areas with frequent fire regimes resulting in changed fuels, or areas with fast paced land use changes. Also, particularly important given the dynamic political situation in the US...

Thank you for this important observation. It is true that the temporal nature of the data is important, but similarly WUI areas change annually, therefore PSEZs are also expected to evolve accordingly, spatially and temporally. Nevertheless, we used the most recent datasets available at the time of the analysis. But to specifically address our concern, a note on data temporality has been added in a new limitations paragraph at the end our revised Results-Discussion section.

Line 159-160: Think about rewording this sentence, as it isn't impervious to fire, it just becomes an urban conflagration not a "wildfire", plus not necessarily low fire occurrence either.

Thanks for this excellent comment. We revised the sentence as suggested and replaced the references with one that are more relevant to WUI-urban fires.

Line 165-167: is this a quote directly from reference 66? I'd suggest just rephrasing and citing.

Done. It now reads, "We therefore used vegetation biomass type and density data—from remote sensing and forest inventories—to better capture the spatiotemporal dynamics of land-cover change processes and how they interact with socio-ecological outcomes."

Line 167: acquisition date for GEDI data?

Thank you for the comment. The acquisition year was already reported in Supplement Table 8, but we have now also included it in the main text to improve clarity.

Line 178: acquisition date for INFC data?

Thank you for pointing this out. The acquisition date was indeed missing in the text and has now been added.

Figure 3: I don't find this figure to be very well formulated. It's maybe just a bit clumsy the way two variables are combined first, and then the third is added. Presenting multivariate data in color has a lot of challenges, but spending some time on getting something presented here that is more intuitive for the reader is strongly recommended. I thought a bit about something to suggest - and I believe a RGB color space cube approach might work for you. You'd have G on the x axis, E on the y axis, and S on the z axis. I've attached a sketch. You could probably keep the existing color scheme (and not have to redo all your maps etc!). Displaying this legend as a 3D shape would be more intuitive than the existing presentation.

Thank you very much for your insightful suggestion. We have now carefully followed both of your recommendations and the draft you kindly provided, and we have revised this figure accordingly.

Line 215: I think Nature Communications keeps results and discussion as two separate sections? Re organization required if so.

As outlined in the Nature Communications guidelines to Authors, both formats—separate Results and Discussion sections or a combined Results and Discussion—are acceptable, depending on the nature of the work. Accordingly, and respectfully, given the methodological nature of our

manuscript, an integrated Results and Discussion section best preserves the narrative flow. By presenting findings alongside the methodological workflow, we can more effectively discuss how each analytical decision shaped subsequent steps in framework construction, and how emerging patterns informed our PSEZs and findings. Given these reasons, we hope you agree that that combining results and discussion into a single, cohesive section is both scientifically justified and enhances clarity for readers. But as previously mentioned, if the Editor requests that they be separated, we will gladly do so.

Line 217-219: *I find this an out of place sentence for the start of the results*

As stated in the previous response, we've structured an integrated Results and Discussion section to keep the narrative flow. We believe that starting with this sentence, will prepare the reader for the following narrative that lays out the integrated methodological-findings rather than expecting specific results and then their discussion in a later section.

Figure 4: I find the maps quite difficult to decipher with so many colors. I don't have any great ideas as an alternative though. Also, if you do re-work Figure 3 as I've suggested above, you could use the cube in place of the legend?

Thank you for your helpful feedback. Figure 4 (a and b) was designed to provide an initial visual overview of the outcome of our method and the resulting Pyro-Socio-Ecological Zones (PSEZ). We experimented with replacing the traditional legend with the revised Figure 3—following your suggestion—to enhance visual coherence. However, this approach required allocating a substantial portion of the figure space to the legend, ultimately compromising the readability of the maps. For this reason, we opted to retain a conventional legend format, while ensuring it remains as clear and concise as possible. We hope you find this acceptable.

Page 13: this section where you're discussing WUI delineation and the limitation that they don't include socioecon/governance factors - as mentioned in my main comments, I'd add a paragraph here that focuses on studies that look at WUI typologies/archetypes or social vulnerability in the WUI.

Thank you again for this comment and suggestion. In addition to our response to your Question #1, we have now included this new paragraph as requested in our revised manuscript in the second paragraph of section 3.1. The sentence now states, "An integrated methodology for mapping the WUI across disparate contexts is crucial to understand management and resource issues associated with global peri-urban areas. There is an extensive body of literature that has developed WUI typologies and archetypes that account for social vulnerability and risk management related to wildfire. For example Schumann et al., (2024)¹, Rivière et al., (2023)⁴, and Paveglio et al., (2015)² studied the social vulnerability, participatory mapping criteria, and adaptive capacity related to wildfires. Similarly, Evers et al., (2019)³ and Wigtil et al., (2016)⁵ have developed archetypes of wildfire exposure and identified places in the USA with high wildfire potential and social vulnerability. However, there are few typologies that account for governance – strictu sensu - in fire-prone WUI zones across disparate contexts. In terms of accounting for the biophysical dimensions, Radeloff et al.'s 2018⁹ pioneering approach of...."

We hope this addresses your concern.

Figure 5: you could consider removing the legend, and just label the zone names directly on the bars in the figure. Would mean the reader wouldn't have to attempt to match colors. I'd probably put the labels between the two bars and just use arrows to point to them.

Thank you for the valuable suggestion. The legend has been removed and the zone names are now directly labeled between the bars with arrows, making the figure easier to read without requiring color matching.

Line 271-272: remove the sentence that starts with "These classes reflect...", not necessary here.

Done.

Line 276-277: not sure "interact more intricately" is the right wording. Its not that they have a more intricate interaction, just more diversity in the categories.

Thanks for the suggestion, we revised the sentence accordingly.

Line 281-282: remove first line, not necessary

Done, the revised text now reads, "Figures 6a and 6b display average annual area burned and number of wildfires for the same PSEZ in both southern California and Italy, respectively. These findings have implications for incorporating socioeconomic and governance dimensions into wildfire management and possibly risk assessments."

Line 293: Figure 6a is Italy though? Also, throughout the manuscript, keep the order of presentation consistent (in text, and in figures) i.e. always do Italy first, or always so Cali first.

Thank you for pointing this out. The typo has been corrected, and the order of presentation has been made consistent throughout the manuscript and figures, with Italy now presented first.

Line 324-327: this sentence is circular, doesn't make sense

We deleted it as part of other revision that were requested by you and other reviewers.

Line 334-336: limitations should be included in discussion somewhere, apologizes if I've missed it there.

Thank you for your comment. Related to the previous comment, we have now included a new paragraph on study limitations at the end of the Discussion, highlighting key issues such as spatial resolution, data comparability, and the temporally dynamic nature of PSEZs.

Line 354: change to "...activities and the effects of global environmental change."?

Thank you for the suggestion. The sentence has been revised as recommended.

Supplementary Material

Supplement table 5: do the authors have any concerns about some of the news sources used for corruption cases?

This is an interesting comment. We understand the importance of using, published peer-reviewed sources for references. And the possibility of bias and accuracy with news outlet reporting on corruption cases. In the case of California (USA), obtaining data for the Transparency indicator proved to be challenging as there were no publicly available Transparency/corruption portals or outlets. So, we conducted a meticulous case-by-case review of all the sources cited in Supplement Table 5 before incorporating the data into the framework. The source 'CalMatters' – which is a publicly available, accessible, site maintained by non-politically appointed functionaries, provides (we felt) an unbiased source. But we acknowledge that more comprehensive data is needed. We

hope that in the future, data on this indicator will be collected, validated, and made available in a more standardized manner in the USA.

It is worth mentioning that Transparency International has been actively working on this issue, and over the years, it has improved its methodology. In addition to the annual 'Global Corruption Barometer,' Transparency International has been refining the 'US Corruption Barometer' annually and this is expected to enhance the quality of the data available for US states, including California.

We have also clearly highlighted the data gaps as a limitation both in the discussion and conclusion sections of the paper, acknowledging that the lack of consistent and comprehensive data is a key limitation in our analysis.

Reviewer #2 (Remarks to the Author):

The premise of this paper is very interesting and compelling. Information about governance would greatly advance spatial data mapping tools indicating the need, potential, and opportunity for wildfire risk reduction in populated fire-prone areas.

We thank the reviewer for the positive and constructive comment.

However, this paper does not provide a comprehensive review of the relationship between governance and its various dimensions and wildfire risk and wildfire risk management. The article also does not describe the data used to create the six indicators, how they were constructed, and precisely what they mean.

The reviewer is correct but we were limited by the fact that the journal has a word limit so there is no way we could provide a comprehensive review of governance dimensions or for that matter fire risk and fire management. An original draft had more introductory content reviewing the wildfire governance literature as well as several references, but we had to remove this section and references to meet these word limits. Also as requested by Reviewer #1, we limited the implications for fire risk and instead focused on the governance implication of fire management in the WUI. With regard to "not describe[ing] the data used to create the six indicators", please refer to Table 1 and Supplement Tables 5 and 6 where you will see a description and references. Again, the journal does not allow us to go into detail on each indicator, but we hope that describing them to the reader and directing them to the source, will suffice.

However we have revised our manuscript in two ways to address your concern. First, we emphasize that our study aim was not to address fire risk and its management. Please see our responses to Reviewer 1's Questions 1 and 13. Second, we have added succinct text and 5 new wildfire governance references in the third paragraph of our Introduction. These sentences read, "Governance is complex, multi-faceted, often theoretical, and can include various aspects such as transparency, participatory processes, trust, and effective and fair institutions"^{10,11}. For examples of studies evaluating, measuring, and analysing for strong or weak wildfire governance structures, please refer to Hamilton et al. (2019)¹², Ager et al. (2017)¹³ and Holm and Fischer (2023)¹⁴.

Again, given the word limitations, we hope this sufficiently addresses your concerns.

The article uses a Human Development Index to capture the "Socioeconomic and demographic dimension," but it is explicitly stated for what purpose, evidently to indicate social vulnerability and the need to reduce exposure?

We thank the reviewer for pointing out the lack of clarity regarding the role of socioeconomic dimension implications of study. We agree that this aspect was previously underexplained.

In response, we have revised the introduction to explicitly articulate how governance realities and socioeconomic factors—such as those captured by the Human Development Index (HDI)—interact to shape resources needs and management capacity in WUI areas (not vulnerability and its components per se). Although some references supporting this perspective were already included, the rationale and conceptual link were not sufficiently developed in the previous version. So to address your concern, we have now added this revised text, “...the Human Development Index (HDI)—interact with governance quality to influence both the institutional capacity to manage wildfires and the levels of public trust, civic engagement, and social cohesion needed to address complex socio-environmental challenges in WUI areas..”

The authors provide more detail about the construction of the pyro and ecological zones. However, how new constructions of these spatial data layers improve upon previous efforts is unclear. The pyro zone seems to be a new name for existing WUI data layers.

This comment is similar to comment #1 by Reviewer 1, so we appreciate the opportunity to further clarify how our PSEZ approach is a contribution and adds a novel perspective on existing WUI typologies. First, while traditional WUI maps—such as those by Radeloff et al. (2005)¹⁵ or various “Wildland Urban Interface” datasets from several countries and regions focus primarily on the spatial juxtaposition of structures relative to vegetation patches (e.g., building density, ember-transport buffers, vegetation cover), these rarely account for the broader socioeconomic or governance contexts that are inherent to WUI systems.

Furthermore, our “Pyro” component is not merely a “relabeling” of WUI vegetation components, since it goes beyond using broad vegetation land cover types and instead uses an index that synthesizes high-resolution vegetation biomass (from GEDI and validated against national forest inventories) with land-cover weighted values. This approach yields a continuous 0–1 metric for flammability potential and ecosystem service provision. This moves beyond static proxies that do not capture actual fuel loads and their spatial variability.

Moreover, by embedding this ecological index alongside separately normalized socioeconomic (HDI-based) and governance (WGI-based) indices we present a single tri-dimensional framework, and 27 unique zone classes that explicitly link vegetation/fuel loads with socioeconomic characteristics of human settlements in the WUI as well as governance measures (i.e., trust, organizational performance, and transparency of participatory processes) in the same WUI system. Previous efforts have either added socioeconomic variables as ancillary layers or secondary analyses; but ours is one of the first to integrate all three dimensions into an integrated spatial typology.

Finally, an additional contribution is the application of our framework in two ecologically similar but administratively and socio-culturally distinct regions—southern California and southern Italy. This demonstrates that WUI is not solely a North American construct, but that a PSEZ approach accounts for different socio-ecological contexts that are different from North American ones. This cross-regional analysis underscores the methodological contribution of PSEZ in facilitating spatially explicit, context-sensitive wildfire management strategies.

Organizationally, the article would have benefited from separating the Results from the Discussion and providing a clear report of results for the overall PSEZs, each zone, and the various indexes. Then, the discussion should focus on the improvements and limitations of these new spatial tools in relation to existing tools and conceptualizations of fire-prone geographies.

This is also similar to a comment from Reviewer 1. As laid out in our study aim, this was a manuscript that proposes and develops a methodological framework that was then applied to two similar ecoregions but with different socioeconomic and governance contexts. Therefore, parsing out the results and discussion would drastically change the aim of the manuscript since our purpose was not to test and present specific objectives and hypotheses, and then discuss findings as done by other studies. But rather- we justify, lay out and apply the PSEZ approach in a repeatable and transparent manner so that it can then be applied for practical purposes across disparate geographies. To separate results from discussion would drastically change our manuscript and would substantially add to the word count; which has limits set by this journal. We hope this explanation is sufficient and justifies our hesitance in splitting up these two sections. But again, if the editor requests it, we will gladly separate them.

Reviewer #3 (Remarks to the Author):

General comments

The topic is interesting....The text is well written with some typos (e.g., than instead of then in line 311; mention to use the US and after use USA)...The relevance of the methodology and the effective impacts of its practical utilization must be emphasized.

Thank you for your careful reading and helpful comments. We have corrected the typo(s) and have now consistently used "USA" throughout the manuscript to ensure uniformity in country references.

Detailed comments

Title

The focus is not only on governance but on wildfire management. Should the title shift to Pyro-1 Socio-Ecological Zones: A Tri-Dimensional Framework for wildfire management in the Wildland-Urban Interface?

*This is an interesting point you make. However, as outlined in our introduction and study aim, the novelty and contribution of our manuscript is the incorporation of governance into mapping and management of fire in the WUI. In fact, WUI by definition usually implies the management (as well as planning and mapping) of wildfire, but rarely its governance. We also note that a Google Scholar search "Wildland-Urban Interface AND management" resulted in 22,800 hits while "Wildland-Urban Interface AND governance" resulted in 14,900 hits. Finally, such a study title, which we cite, already exists (Jenerette, G. Darrel, et al. "An expanded framework for wildland-urban interfaces and their management." *Frontiers in Ecology and the Environment* 20.9 (2022): 516-523.). So, given this, we would rather leave the title as is. However, to meet your suggestion half way, we have now included "wildfire management" as a key word. We hope this is acceptable to you.*

Abstract

Line 12, you did not consider the ecological dimension and inserted the management that is not considered in the framework.

We thank the reviewer for noticing this oversight. It was a typo, and we have promptly corrected.

In the keywords, you consider Land use change and Land cover change, although you just considered land cover in the framework. In addition, there is no mention of the temporal shift in land cover. What is the relevance of considering peri-urban areas?

Thank you for your observation. You are correct in noting that the framework addresses land cover change but not land use change. We have therefore removed "land use change" from the list of keywords and replaced it with "wildfire management" as also suggested by Reviewer #1. We also acknowledge the comment regarding the lack of mention of temporal shifts in land cover. As this study is based on a static assessment of land cover, temporal dynamics were not considered and fall outside the scope of the current framework.

(What is the relevance of considering peri-urban areas?)

We have added the relevance of PSEZ in the abstract and in our future research section at the very end of our Results-Discussion section. Specially the new text reads, "...Second, our PSEZ approach was used for the WUI and fire, but it could also be adapted and applied to peri-urban areas where issues such as ecosystem service provision, conserving habitat, and formulating land use policies near human settlements are more relevant than fire..."

Introduction

How do you define WUI and peri-urban areas? How did you spatially define the WUI area? And the peri-urban areas? How did you delimit the WUI?

We have word limits, but these two concepts are succinctly defined and differentiated in the first 2 sentences of the introductory paragraph. But basically, "peri-urban" is the general term that described the spatial transition between urban and rural areas and "encapsulates [all] the challenges of urbanization, ecological, and environmental impacts". WUI on the other hand is strictly wildfire related and in the Third paragraph of our Results and Discussion we describe exactly what WUI is and how it is mapped. In terms of how we mapped the WUI in our study, please refer to Methods Section 2.3.1.

Line 62-67 - This statement is interesting. You must demonstrate how your framework can be used to enhance wildfire management effectiveness, improve governance, build resilience, and adaptive capacity.

Thank you for this insightful suggestion. We agree that demonstrating direct applications of the framework—in terms of operational wildfire management, governance improvement, and resilience-building—would be highly valuable. However, such empirical validation and operational implementation are beyond the scope of our study. Here, our primary goal was to develop and rigorously test a novel tri-dimensional classification of, and mapping framework for, the Wildland-Urban Interface via PSEZs to better inform scholars and decision-makers with a more nuanced spatial typology. By providing a standardized, context-relevant mapping framework, we lay the necessary groundwork for subsequent research and projects to do exactly what you state- identify zones for: (effective) targeted management actions, (improved) policy instruments, and enhanced the adaptive capacity of socio-ecological systems.

Methodology

What is the dimension of the pixel used in the maps? For biomass, the spatial resolution of 1 x 1 kilometre

We thank the reviewer for the helpful comment. Indeed, the maps of the three indices — Socioeconomic (S), Governance (G), and Ecological (E) — were not generated using a fixed pixel size and spatial resolution, but rather through a spatial overlay and intersection of vector-based datasets. More specifically, each index was calculated using the smallest available administrative

or statistical units (e.g., municipalities or census tracts), and then mapped by spatially intersecting these polygon-based units with the relevant input layers (such as land cover, biomass, or socio-demographic data). This process was performed in a GIS environment, ensuring that each resulting map reflects the combination of data values aggregated or assigned within each spatial unit, rather than being based on a uniform raster resolution.

In response to the reviewer's comment, we have now clarified this methodological detail in the manuscript, specifying that the maps are the result of vector-based spatial intersections, not raster pixel-based calculations.

What is the relevance of each of the variables selected?

Thank you for your comment. The variables selected in our Pyro-Socio-Ecological Zones (PSEZ) framework are grouped into three main dimensions—Ecological, Socioeconomic, and Governance, each of which plays a critical role in characterizing wildfire-prone Wildland-Urban Interface (WUI) zones. Their relevance is explained as follows:

1. Ecological variables:

These indicators capture environmental and landscape-level features that influence wildfire occurrence and spread. These variables are essential because they affect biomass/fuel availability and flammability, directly influencing fire behavior and ecosystem vulnerability.

2. Socioeconomic variables:

These variables reflect the demographic and socioeconomic characteristics of the human communities residing in WUI zones. Their relevance is in accounting for human presence, the socio-demographics characteristics of humans in the PSEZS, different community and actor groups, possible drivers affecting ignition risks, and implication for the vulnerability of different demographics to wildfire.

3. Governance variables:

Governance indicators measure institutional performance, transparency, trust, and public participation mechanisms. This dimension is particularly novel in WUI studies as explained in our first response to you. Its inclusion allows us to account for the enabling conditions that influence the implementation and effectiveness of fire management policies, land-use planning, and community resilience strategies.

So, by combining ecological, socioeconomic, and governance dimensions, the PSEZ framework provides an integrated and spatially explicit understanding of wildfire management and the implication to wildfire risk and vulnerability.

How can the framework help to identify management needs and areas of vulnerability?

Done. As requested by a previous reviewer, we now address this in our future research section at the end of our Results-Discussion section. Specifically the new text reads, "That said, our method and approach could be used for future research. First, the data and methods could be used to develop a fourth dimension like vulnerability or wildfire risk management (12,34). Although fire risk has been studied and methods have mapped it (4, 9, 34,75), governance is different than vulnerability, however most of our socioeconomic and governance indicators could easily be incorporated into a framework to account for the exposure, sensitivity and adaptive capacity components of vulnerability²⁹."

How can you obtain Gross National Income (GNI) per capita values adjusted for purchasing power parity at the pixel level?

We thank the reviewer for this valuable comment, but HDI values in our study were not estimated at the pixel level, but rather based on the smallest administrative/statistical units for which official data are available. As reported in "Supplement Table 4. Scale of HDI Estimation for Italy and California", these are: Municipalities in Italy (ranging from 0.07 to 590.99 km²) and U.S. Census Tracts in California (ranging from 0.06 to 3024.62 km²).

Additionally, we have now revised the explanation of the sources and methods we used for estimating the Human Development Index (HDI) in both Italy and California. Specifically, regarding Gross National Income (GNI) per capita adjusted for purchasing power parity (PPP), we clarify that for Italy, this data was obtained from the Italian National Institute of Statistics (ISTAT), which provides standardized economic indicators at the municipal level. For California, as clarified in the revised manuscript, the HDI values—including the income component—were taken directly from the dataset developed by the Social Science Research Council's Measure of America project, which offers pre-calculated HDI scores at the U.S. Census Tract level.

In the pyro component of the model, you use WUI. WUI is the area where you are going to apply the model. Thus, it should be out of the model. In the pyro component, wildfire characteristics could be another option, or remove pyro from the model as a component.

Thank you for your comment. Following your suggestion, we have removed "WUI" which was previously included as the pyro component of the model—from the model inputs. Instead, we now clearly present the WUI as the spatial and conceptual context within our tri-dimensional framework, rather than as a component of the framework itself. This also aligns with your subsequent comment regarding Figure 2. Accordingly, we revised the figure and developed a new flowchart that better illustrates the structure and logic of the framework.

How can you get the data of the governance indicators at the pixel level?

We appreciate the reviewer's observation. But unlike our ecological dimension, governance and socio-economic are social phenomena that occur at difference scales than plant communities or land cover types. So, HDI and governance indicators were not derived at the pixel level, but based on municipality-level data, which represent the most detailed spatial units for which governance-related variables are officially available in both Italy and California.

This comment also helped us identify an inaccuracy in our manuscript: unlike other indicators, the spatial resolution of our governance indicator had not been explicitly specified. To address this, we have now added a new table — Supplement Table 5. Scale of Governance Indicators (G) Estimation for Italy and California — which clarifies the spatial scale used:

- Italy: Municipalities ranging from 0.07 to 590.99 km²*
- California: Municipalities ranging from 0.08 to 1302.06 km²*

This update ensures consistency and transparency across all variable descriptions.

The weight attributed to the indicator should be considered in a table in the text

Thank you for the suggestion. The assigning of equal weights to each indicator has been clarified in the manuscript text. The last sentence of the first paragraph in Section 2.4 now reads, "Each index was considered equally important in the analysis and therefore were assigned equal weights in the classification process."

In Figure 2, the three dimensions are pyro, socio, and ecological; however, in the text, you mention S, G, and E. What about removing pyro and separating socio into socio and governance?

We apologize for this confusion. The 3 dimensions are socioeconomic (S), ecological (E) and governance (G). So, we now revise the figure by separating socio into social and governance, in line with the S, G, and E dimensions mentioned in the text.

Results and discussion

You state "Our integrated PSEZ framework facilitates more informed and targeted land management actions and policies aimed at mitigating wildfire risks and enhancing landscape resilience and ecosystem services". Please justify how this can be obtained; otherwise, it is a general statement.

It is indeed meant to be a general statement. And this is why we purposefully placed it at the very beginning of our Results and Discussion section. As you will read in the following text and sections, we believe we make the case for how this framework and our proposed method can facilitate these goals. Please note that in our revised future research section and conclusion we specifically address these issues you bring up.

Line 308 - What is the unit of the value (2032.38)?

We thank the reviewer for the observation. The unit has been clarified in the manuscript as "2032.38 events in total for the considered period," and the text has been revised accordingly.

You state that "the PSEZ framework can be useful in capturing the interactions among ecological, socioeconomic, and governance factors that influence wildfire dynamics across disparate contexts." In my point of view, you do not analyse the interactions between ecological, socioeconomic, and governance factors.

You are correct in that we do not 'quantitatively analyze' the interactions. Rather we graphically map and display them, and in doing so we better understand these interactions among the 3 dimensions- We hope you agree with this. So to better reflect the aim of our study and framework, and your comment, we revised this sentence and 1-2 others that infer that we are 'analyzing' or 'examining' interactions by replacing "analyzing/examining" interaction with "better understanding" the interactions among the 3 dimensions. We hope this is acceptable.

You state " Its application in southern California and southern Italy shows that socio-ecological and governance dynamics in disparate WUIs are shaped by both ecological dimension, local governance capacities, and socioeconomic factors." In my point of view, this is a round explanation. It is the first time you mention local governance capacities. What is the meaning?

We modified the entire paragraph. The revised text now reads, "Its application in southern California and southern Italy shows that socio-ecological and governance dynamics in disparate WUIs are indeed shaped by complex ecologies, varying levels of governance capacities, and diverse socioeconomic factors."

Conclusion

Emphasize clearly the strengths and limitations of the work.

Please see the last 2 paragraphs in the previous revised Results and Discussion section. As requested by other reviewers as well, we now specifically mention our limitations and areas of future research. The new paragraphs read, "However, this study has some limitations. First, the

availability, quality, and spatial resolution of the datasets will vary considerably across regions, often reflecting differences in methodologies adopted by national and regional governments. In particular, the spatial resolution of socioeconomic and governance data may fail to capture fine-scale heterogeneity within peri-urban or WUI landscapes since the size and configuration of administrative units (e.g., municipalities, census tracts) will differ within and between countries. Second, PSEZs are dynamic and will change over time in response to ecological changes, socioeconomic developments, and shifts in governance structures thus there is a need for long-term monitoring and updates to ensure relevance and accuracy.....That said, our method and approach could be used for future research. First, the data and methods could be used to develop a fourth dimension like vulnerability or wildfire risk management (12,34). Although fire risk has been studied and methods have mapped it (4, 9, 34,75), governance is different than vulnerability, however most of our socioeconomic and governance indicators could easily be incorporated into a framework to account for the exposure, sensitivity and adaptive capacity components of vulnerability...”

References used in the above responses:

1. Schumann, R. L. *et al.* The geography of social vulnerability and wildfire occurrence (1984–2018) in the conterminous USA. *Nat Hazards* **120**, 4297–4327 (2024).
2. Paveglio, T. B. *et al.* Categorizing the Social Context of the Wildland Urban Interface: Adaptive Capacity for Wildfire and Community “Archetypes”. *Forest Science* **61**, 298–310 (2015).
3. Evers, C. R., Ager, A. A., Nielsen-Pincus, M., Palaiologou, P. & Bunzel, K. Archetypes of community wildfire exposure from national forests of the western US. *Landscape and Urban Planning* **182**, 55–66 (2019).
4. Rivière, M., Lenglet, J., Noirault, A., Pimont, F. & Dupuy, J.-L. Mapping territorial vulnerability to wildfires: A participative multi-criteria analysis. *Forest Ecology and Management* **539**, 121014 (2023).
5. Wigtil, G. *et al.* Places where wildfire potential and social vulnerability coincide in the coterminous United States. *Int. J. Wildland Fire* **25**, 896 (2016).
6. D’Este, M., Giannico, V., Laforteza, R., Sanesi, G. & Elia, M. The wildland-urban interface map of Italy: A nationwide dataset for wildfire risk management. *Data in Brief* **38**, 107427 (2021).
7. Lambrou, N., Kolden, C., Loukaitou-Sideris, A., Anjum, E. & Acey, C. Social drivers of vulnerability to wildfire disasters: A review of the literature. *Landscape and Urban Planning* **237**, 104797 (2023).
8. Fischer, A. P. *et al.* Wildfire risk as a socioecological pathology. *Frontiers in Ecol & Environ* **14**, 276–284 (2016).
9. Radeloff, V. C. *et al.* Rapid growth of the US wildland-urban interface raises wildfire risk. *Proc. Natl. Acad. Sci. U.S.A.* **115**, 3314–3319 (2018).
10. Nikolakis, W. & Roberts, E. Wildfire governance in a changing world: Insights for policy learning and policy transfer. *Risk Hazard & Crisis Pub Pol* **13**, 144–164 (2022).
11. Aguilar, S. & Montiel, C. The challenge of applying governance and sustainable development to wildland fire management in Southern Europe. *Journal of Forestry Research* **22**, 627–639 (2011).
12. Hamilton, M., Fischer, A. P. & Ager, A. A social-ecological network approach for understanding wildfire risk governance. *Global Environmental Change* **54**, 113–123 (2019).
13. Ager, A. A. *et al.* Network analysis of wildfire transmission and implications for risk governance. *PLoS ONE* **12**, e0172867 (2017).
14. Holm, F. & Fischer, A. P. Combining multiple data sources to identify actor involvement in environmental governance: Wildfire in the American West. *Environmental Science & Policy* **147**, 361–378 (2023).
15. Radeloff, V. C. *Et al.* The wildland–urban interface in the united states. *Ecological Applications* **15**, 799–805 (2005).

REVIEWERS' COMMENTS

Reviewer #1 (Remarks to the Author):

The authors were very detailed and mindful in their revision and responses to the review comments. Excellent job, it's a really nice paper. I think this manuscript is ready to go, short of a few very minor edits. I've got some noted below, by line number:

Thank you for your detailed and helpful comments. They really improved the manuscript.

Line 30: WUI, not WUIs

The correction has been made. Thank you.

Line 32: throughout paper, remove the extra spaces before the superscript references and make sure format matches that of the journal requirements

The corrections have been made throughout the manuscript, and we have checked for formatting as also requested by the Editors. Thank you.

Line 50: WUI, not WUIs

Done. Thank you.

Line 72: Maybe change to "Development of such a framework and approach could ultimately help..."

We have accepted your suggestion and made the changes as requested. Thank you.

Line 83-87: flip order so it is always ITA then Cali

Done. Thank you.

Figure 2: use full name ("Socioeconomic") on the box currently labeled just "Socio" to match the other 2 dimensions. Figure is much improved from last version

The figure has been revised following this latest suggestion. Thank you.

Line 118: the subsection heading 2.3.2.1 is probably not necessary?

This has now been deleted since the journal does not permit more than 1 level of subheadings.

Line 119: "and demographic" included in S? If it is socioecon and demographic, then that needs to change in Fig 2 and in text above where S is referred to as socioecon

Done.

Line 123: “cross-country” is a bit ambiguous, maybe use “intra-comparison of countries”?
We have accepted your suggestion and made the changes as requested. Thank you.

Line 136: remove link here, just cite in references

Done.

Line 139: this section should be 2.3.3 I think? Then adjust following sections accordingly
The subsection has been deleted due to Editor request as explained previously.

Line 140: “Governance plays a crucial role in managing complex issues in the WUI40. Therefore” is not necessary here. Start sentence with “Governance, our second...”. If you’d like to include this point, maybe move to intro where governance is discussed?

This is an interesting comment, and we see your point since not including this sentence would indeed make the paragraph more succinct. Interestingly, this sentence was not included in our original submitted version. However, because Reviewer 2 requested this sentence, in the previous revised version we included it in this last revision. Therefore, we probably should keep it as is.

Line 143: Why is ref #25 not a reference to the source data for the WGI? Also look at formatting for the ref here, maybe try “(WGI; Supplement Table 5)25.”

We thank the reviewer for catching this oversight. The incorrect citation was a typo, but has now been corrected. In addition, the reference format has been revised according to the reviewer’s suggestion.

Line 143-145: you probably do not need to include the original list of indicators from WGI, and could get away with just saying that your indicators are based on theirs, but modified based on what data you had access to that could represent those indicators? Or any other justification on their selection. I was confused looking at the list here vs the table that they didn’t match and had to re-read for clarity. And I’m not sure why ref #41 is here, it doesn’t mention any of these indicators.

We thank the reviewer for this suggestion. However, Reviewer 2 is specifically requesting more explanation regarding our Governance index, so explaining the WGI framework is necessary. However to clarify the text we have modified the second half of the paragraph and it now reads, “The WGI has been used in over 200 countries to account for societal perception of governance quality and is based on six aggregated indicators: Voice and accountability, Political stability

and absence of violence/terrorism, Government effectiveness, Environmental regulatory quality, Rule of law, and Corruption control 25,41. Accordingly, we adapted this framework based on data availability and their socio-ecological relevance to the two WUI contexts we studied as shown in Table (1) and Supplement Table (6).” In addition, reference #41 has been replaced with a more pertinent citation: Thomas, M. A. What Do the Worldwide Governance Indicators Measure? Eur J Dev Res 22, 31–54 (2010).

Line 150: change numbering of section to 2.3.4, and I’d suggest re-naming the section to “Ecological Index” and then move lines 192-199 here (with some slight re-wording for flow). Then use the subheadings for “Land Cover” and “Vegetation biomass” to follow the paragraph on the ecological index. I think that would flow better, and also match the structure of the sections for the other indices.

Thanks for the suggestion. We have moved the paragraphs as requested but as previously explained we have had to delete all second and third level subheadings.

Figure 3: yes glad this presentation worked out, looks good!

Thank you.

Line 235: change “highlights” to “highlight”

Done.

Line 284: remove “The”

Done.

Fig 5: heading at the top of the figure image is not necessary, remove and rely on the figure caption below.

The figure heading has been improved per your suggestion. Thank you.

Line 293-300: I just realized it would be nice to add in some descriptive statistics on the diversity to go along with your description here. Maybe a pure count of how many zones each region has (i.e. “richness”), and then maybe something like the Simpson’s index, or relative diversity/evenness?

We sincerely thank the reviewer for this valuable suggestion. We fully recognize the potential relevance of descriptive statistics such as richness or diversity indices (e.g., Simpson's

index, evenness) in similar contexts. However, we have chosen not to include them in this case, as introducing an additional set of analyses would require changes to the Materials and Methods, Results, and Discussion sections, potentially distracting from our study aim and word count limits. Further, our study is already based on three composite indices, and adding further diversity metrics could overwhelm the reader with information, add complexity to the manuscript's structure and reduce clarity for the reader, rather than reinforcing the study aim.

However to address your observation on PSEZ diversity, we have revised selected text that refers to Figure 5. For example, in the sentence preceding Figure 5 it now states, “ Figure 5 illustrates the variability in both the distribution and diversity in PSEZs between southern California and southern Italy.” The paragraph following Figure 5 was revised and now reads, “Specifically in southern California, the ecoregions are characterized by 24 PSEZs, two of which dominate (i.e., EL-SM-GH and EL-SH-GM), and together cover a significant portion of the study area (52.76%). In contrast, southern Italy exhibits much more fragmented and diversified ecoregions dominated by 27 PSEZ. The PSEZs with the largest area in southern Italy are characterized by EM-SM-GH (15.26%), EH-SM-GH (14.49%) and EL-SM-GH (9.06%). Thus, the greater array of PSEZ dimensions in Italy reflect a complex landscape characterized by greater diversity in terms of socioeconomic and governance realities (Figure 5). This variability points to a mosaic of territories and landscapes with diverse human communities and management needs, emphasizing the heterogeneity of the socio-ecological context of southern Italy^{85,86}.” We hope this addresses your comment.

Line 302: change “for the same” to just “by”

Done.

Line 325: change to “Conversely, southern California...”

Done.

Line 330: change to “This result suggests that...”

Done.

Line 339: the start of this paragraph doesn't flow from the previous

Thank you for this suggestion, we have revised the beginning of the sentence and it now reads, "More specifically, southern Italy's realities differ from those of southern California, where....."

Line 358: reference format needs adjusting

Done- It must have been a typo.

Line 377: change "in" to "identify"

Done. Thank you.

Line 378-379: maybe change to "...are at risk. This knowledge could be used to target attention on more effective..."

The correction has been made. Thank you.

Line 382: change "like" to "such as"

Done.

Line 383-384: not sure the wording here is right. Maybe merge the point with your point at the end of the paragraph?

We have revised this sentence, and it now reads, "Accordingly, our PSEZ underscore the need for well-defined management strategies in peri-urban areas and contexts that also align with global sustainability goals."

Supplementary materials: I'd suggest that anywhere possible, try to condense text in the tables, use point form (e.g. the descriptions in Table 1)

Thank you for your suggestion. But to more effectively address comments from Reviewer 2, we prefer to leave the text and table as is to better address their requests regarding more detail and explanation of our different indices.

Suppl Line 2: capitalize "Socio"

Done.

Reviewer #2 (Remarks to the Author):

The authors made some effort to revise the paper to address the reviewers' comments. However, the changes were limited. The paper still does not provide a coherent conceptual model of governance and how it relates to wildfire risk or how it could be managed more effectively.

We thank Reviewer #2 for the additional feedback and time taken in reviewing our manuscript. We acknowledge the relevance of the points they raised concerning the conceptual grounding of governance, the role of the HDI, and the novelty of the ecological index. These are indeed important aspects, and we agree that both an extensive theoretical piece or a “United Nation-Food and Agriculture Organization type” fire management technical/policy manual with guidelines on how to implement our PSEZ framework is warranted. However, this was not the aim of the manuscript, nor is this request within the scope of this journal we feel. That said, we would like to let you know that we went through significant efforts to address your concerns as best as possible, but within the constraints of the journal’s editorial guidelines and other reviewer requests. We note that although Nature Communication has no word count limits in the Methods and Supplementary Materials sections, this is not the case in the other sections where your comments and requests apply.

Thus, the scope of the journal, these strict word and reference limits, and specific requests regarding succinct presentation and methodological explanation regarding our ecological, socioeconomic and governance dimensions by the other 2 reviewers, required us to eliminate or condense sections and text that in earlier drafts of the manuscript contained a detailed review of the relevant literature, methodology for development of our governance index, discussion of findings and recommendations. In fact, our original manuscript dedicated 2 paragraphs in the Introduction, 3 paragraphs in the Methods, 3 paragraphs in the Results and Discussion to the state of the art and background on wildfire-related governance. Of the original 100+ references in our first version, nearly 25% of them alone contained the term ‘governance’ in the title. So, we had to strike a careful balance between providing sufficient explanation and ensuring compliance with these requirements.

That said, please refer to the following sections to see how we have attempted to address your concerns in this revision with succinct new material in both the main and supplementary texts.

While the indicators intuitively seem important, the authors do not explain their relevance. The authors did little to address my previous comments about this shortcoming, and the few sentences they added simply state that the constructs are important, not why.

To address this and other Reviewers’ comments, we opted in the previous revision to make targeted clarifications and to strengthen key sections by referencing relevant studies, rather than

extending the text further. This approach allowed us to balance the amount of text equally among the governance, socioeconomic, and ecological dimensions, while remaining within the editorial limits. In fact, as the Reviewer noted in the first round, our rebuttal provided a more extensive discussion of these conceptual underpinnings; unfortunately, it was not possible to reproduce that level of detail in the main manuscript that the reviewer is requesting.

However, to address your concern we have revised the relevant section and succinctly explained the WGI indicators in the main text of our methods section. The revised Governance Index section now reads, “Governance plays a crucial role in managing complex issues in the WUI⁴⁰. Therefore governance, our second dimension was accounted for by using the composite Governance Index (G hereafter), ranging from 0 to 1, and that was based on the 2024 Worldwide Governance Indicators (WGI; Supplement Table 5)⁴¹. 25,41. The WGI has been used in over 200 countries to account for societal perception of governance quality and is based on six aggregated indicators: Voice and accountability, Political stability and absence of violence/terrorism, Government effectiveness, Environmental regulatory quality, Rule of law, and Corruption control ^{25,41}. Accordingly, we adapted this framework into 6 indicators based on data availability and their socio-ecological relevance to the two WUI contexts we studied as shown in Table (1) and Supplement Table (6), and as justified in Supplementary Methods Section 1.2.”

Furthermore, in our Supplementary material section we now include a new section “1.2. Justification of governance index methods” and another 15 mostly new references where we now justify our Governance and WGI indicators and their relevance to our 2 study areas. The two paragraphs read, “Our study areas in southern Italy and California face unique challenges due to their geographical and socioeconomic conditions and as a result, require context-specific governance strategies^{40,41}. Most importantly, given their Mediterranean climate, these landscapes are especially susceptible to wildfire risk because of arid, hot summers, their biodiversity, and many of the same anthropogenic pressures and extreme climate event-related challenges⁴². However, applying the same and existing WUI typologies to southern Italy and California⁴³ fails to capture their unique socio-political and cultural diversity and management challenges related to fire risk⁴⁴. More specifically, both regions experience similar urban development and environmental preservation issues, but they differ in governance frameworks and socio-ecological contexts^{45–47}. For example, California relies on federal guidelines and

localized assessments to guide fire risk mapping strategies, whereas Italy's decentralized governance results in a range of regional definitions and regulations⁴⁸. Billings et al. (2021)⁴⁹ also found that trust in government influenced resident support for incentives and aid in implementing fire risk mitigation measures in the WUI.

Therefore, spatially explicit indices and mapping approaches could assist in designing more effective conservation, strong governance, and sustainable stewardship practices^{50,51}. But socioeconomic and governance indicators need to be adapted to reflect localized contexts and conditions particularly at the scale of human settlements such as communities and municipalities^{52,53} Accordingly, we adapted the Human Development Index⁵⁴ and the Worldwide Governance Indicators⁵⁵ to our framework. Both indices have been used internationally across different socio-political scales by organizations such as The United Nations and The World Bank to account for not only human development and wellbeing but also experts and citizen perception regarding the quality of governance. Section 2.3 in the main text details how they were both adapted into our two study areas and PSEZ framework.” We hope you find these revisions acceptable.

Additionally, the authors statement that the study did not aim to address fire risk and its management was perplexing. At the end of the introduction, the paper explicitly states that the framework will help policymakers implement more effective fire management and governance strategies. Incidentally, the paper never explains how the framework will do this. Revisiting the claim about this implication would have been helpful in the discussion.

Thank you for this comment. But, as stated in the last paragraph of our Introduction, our study aim was to “...develop a spatially explicit socio-ecological mapping approach and typology to account for governance in fire-prone WUI zones.” That is what the manuscript does we feel. So yes, now that it has been developed it could be used to “enhance the resilience and adaptive capacity of these PSEZ against wildfire risk”. But we would argue that it can also be used to:

- *“prioritize conservation efforts”*
- *“implement more effective fire management and governance strategies”*
- *“help in developing strategies that safeguard both ecosystems and human communities across disparate regions”*

- *“help enhance the resilience and adaptive capacity of these PSEZ against ... socio-ecological impacts”, and*
- *“facilitate more effective, and equitable strategies for peri-urban management”*

So, as you can now see, attempting to discuss and explain all its potential application is a herculean task so we can think of no way of how to succinctly explain “how the framework will do [ALL] this”? Therefore, we think this is outside the scope of this manuscript.

Again, as previously explained, we had word count limits and requests by the other 2 Reviewers for reducing our text. But more importantly, we believe that this request is, again, not possible since this is a scientific manuscript- it is not a management plan or policy document that outlines fire management across 25-27 unique PSEZs distributed across 4 different ecoregion and 2 different countries.

However, to address your comment regarding “Revisiting the claim about this [fire risk management and governance strategies] implication, we have revised and added text to the fourth paragraph of our “3.2. Wildfire in the PSEZ” section. The paragraph now reads, “A discernible trend in the number of wildfires shows that PSEZs with high governance (GH) consistently present a lower number of wildfires compared to PSEZs with low governance (GL) values. Also, PSEZs with medium to low governance and socio-economic values, show greater wildfire occurrence and burned area, likely due to resource scarcity and a weak governance capacity to manage wildfire in these landscapes. Similarly, cohesive social structures in less fragmented montane regions often result in more integrated and effective community efforts towards resource and management¹⁸. Thus, greater governance index values in Italy’s South Apennine mixed montane forest ecoregion might stem from more effective policy implementation and enforcement where less populated, possibly more cohesive populations, have strong local governance structures. Using this same ecoregion or Southern California Mountains as an example, the S index values might also imply improved public services and economic stability (Figure 3 and 4).”

We hope this addresses your concerns.

Again, the article uses a Human Development Index to capture the "Socioeconomic and demographic dimension," but the authors do not explain the relevance of the indicators to the prioritization of conservation efforts or the implementation of more effective fire management

and governance strategies to safeguard both ecosystems and human communities. The added sentence about socioeconomic conditions interacting with governance quality was general and somewhat vague....I don't see any edits to the section on socio-eco-demographic characteristics, so it appears that the authors were not able to provide a clear explanation of the relationship between the HDI indicators and governance and wildfires.

Please see our previous 2-3 responses; we feel that they can succinctly answer your comment.

The authors still do not explain how the ecological index improves upon existing spatial wildfire risk data products. While the authors provided an interesting discussion of the novelty of their framework in their rebuttal document, I see very few edits to that section of the paper.

Incorporating some of the rebuttal comments into the revised manuscript would have helped.

Thank you for this comment, but please see our responses to your first 3 comments for an answer.

While the authors tried to address the limitations of their study, they did not expand the discussion of the findings or how their framework could improve upon existing tools and conceptualizations of fire-prone geographies. They could have discussed questions such as: What are the implications of the various combinations of governance, HDI, and ecological index values for wildfire risk, and fire management and governance strategies? Why would various levels of the governance indicators be associated with the number of wildfires and burned area?

Thank you for this comment. But again, asking us to do this across 25-27 unique PSEZs distributed across 4 different ecoregions and 2 different countries would be a herculean task plus at the end of the day, would simply not be allowed by the journal. So we maintain that this revised version of the manuscript that incorporates Reviewer 1's and the Editor's specific comments – as well as yours - and the new revised version, reflects the maximum level of conceptual and empirical clarification that can reasonably be included. Accordingly, we believe the framework, as presented, offers a clear and policy-relevant contribution, while also pointing to areas where future studies can expand on both the applied and the theoretical aspects raised by the Reviewer.

In fact, some of the additions to the Discussion introduced more questions than answers. How could the framework be used to "develop a fourth dimension like vulnerability or wildfire risk management"? How could the socioeconomic and governance indicators account for wildfire exposure?

Unfortunately, this is not possible given our previous explanations and given the journal's aims, scope and editorial requirement. We did discuss this and we concluded that there is no way we can address this given the socio-ecological heterogeneity of our 2 study areas. As shown in our manuscript, we identified 50+ different PSEZs across our 4 ecoregions and 2 countries: each PSEZ with its own distinctive governments and socio-ecological communities. Ideally, to address your comments, we would need a series of context-relevant extensive technical documents with specific management, planning and policy guidelines unique to each PSEZ. And in fact it is something we are exploring to do ones this is published,

But, as explained earlier in Section 3.2, fourth paragraph we did add new text in the hope we partially address your concerns that states, "Similarly, cohesive social structures in less fragmented montane regions often result in more integrated and effective community efforts towards resource and management¹⁸. Thus, greater governance index values in Italy's South Apennine mixed montane forest ecoregion might stem from more effective policy implementation and enforcement where less populated, possibly more cohesive populations, have strong local governance structures. Using this same ecoregion or Southern California Mountains as an example, the S index values might also imply improved public services and economic stability (Figure 3 and 4)."

Then in the second to last paragraph we revise the last 2 sentences to better elaborate on the implication of our findings. The revised sentences read, "Similarly, our PSEZ approach could be useful to identify where fire risk might increase due to greater accumulations of flammable biomass and human-induced ignitions but can also identify areas where governance systems are weak, communities lack resources, and infrastructure are at risk. This knowledge could be used to target attention on more effective cross-regional public messaging and communications campaigns, fire suppression efforts and fuel management strategies."

We hope this addresses your concerns.

Reviewer #3 (Remarks to the Author):

Dear authors, Thank you for your replies.

Thank you for taking the time to review our manuscript and for your helpful suggestions.